# Probing the segregation of evoked and spontaneous neurotransmission via photobleaching and recovery of a fluorescent glutamate sensor

**Camille S Wang[1], Natali L Chanaday[2], Lisa M Monteggia[1,2]\*, Ege T Kavalali[1,2]\***

[1]Vanderbilt Brain Institute, Vanderbilt University, Nashville, United States;
[2]Department of Pharmacology, Vanderbilt University, Nashville, United States

**Abstract** Synapses maintain both action potential-evoked and spontaneous neurotransmitter release; however, organization of these two forms of release within an individual synapse remains unclear. Here, we used photobleaching properties of iGluSnFR, a fluorescent probe that detects glutamate, to investigate the subsynaptic organization of evoked and spontaneous release in primary hippocampal cultures. In nonneuronal cells and neuronal dendrites, iGluSnFR fluorescence is intensely photobleached and recovers via diffusion of nonphotobleached probes with a time constant of ~10 s. After photobleaching, while evoked iGluSnFR events could be rapidly suppressed, their recovery required several hours. In contrast, iGluSnFR responses to spontaneous release were comparatively resilient to photobleaching, unless the complete pool of iGluSnFR was activated by glutamate perfusion. This differential effect of photobleaching on different modes of neurotransmission is consistent with a subsynaptic organization where sites of evoked glutamate release are clustered and corresponding iGluSnFR probes are diffusion restricted, while spontaneous release sites are broadly spread across a synapse with readily diffusible iGluSnFR probes.

**\*For correspondence:**
lisa.monteggia@vanderbilt.edu
(LMM);
ege.kavalali@vanderbilt.edu
(ETK)

## Editor's evaluation

Chemical neurotransmission is the major form of inter-neuronal communication in the CNS. The classical model is that the pathways utilized by action potential-evoked and spontaneous neurotransmission are the same. Multiple lines of evidence now suggest that the Venn diagrams describing the vesicle identity, regulatory systems, and postsynaptic receptors utilized by evoked and spontaneous transmission, overlap incompletely. In this paper, Wang and colleagues, explore the distribution of glutamate release sites at hippocampal synapses using a genetically encoded glutamate sensor and photobleaching, and their results indicate that evoked release is confined to a smaller region of each excitatory synapse than the more dispersed spontaneous release.

## Introduction

Synapses form the basis of neural communication and are exquisitely organized in facilitating point-to-point neurotransmission. Neurotransmitter release can be broadly organized into two main categories – evoked and spontaneous release. Evoked release is the more studied form of release and occurs in response to an action potential (*Südhof, 2013*). Spontaneous release, also referred to as activity independent release, occurs independently of action potentials. Several recent studies suggest that evoked and spontaneous release have functional segregation. For example, spontaneous and evoked release derive presynaptically from distinct synaptic vesicle pools (*Sara et al., 2005*), and use different

release machinery as well as activate nonoverlapping sets of postsynaptic receptors (*Atasoy et al., 2008*; *Farsi et al., 2021*; *Kavalali, 2015*; *Melom et al., 2013*; *Peled et al., 2014*; *Sara et al., 2011*). Moreover, spontaneous release can activate distinct postsynaptic downstream signaling pathways compared to evoked release (*Horvath et al., 2021*). Spontaneous release has also been implicated in mechanisms underlying neuropsychiatric and neurological diseases as well as their treatments separate from evoked release (*Alten et al., 2021*; *Autry et al., 2011*; *Kavalali and Monteggia, 2012*).

While studies have proposed a functional segregation of the two forms of release, the exact subsynaptic organization supporting spatial segregation remains unclear. Electrophysiology experiments utilizing use-dependent blockers to examine the segregation of postsynaptic receptors provide high temporal resolution of receptor activation, but do not contain spatial information about their subsynaptic location (*Atasoy et al., 2008*; *Horvath et al., 2020*). Optical imaging of fluorescent neurotransmitter probes has the advantage of reporting where neurotransmitter release occurs. For instance, wide field fluorescence imaging of fluorescent probes has examined organization of spontaneous and evoked release across different synapses (*Reese and Kavalali, 2016*), though it could not resolve subsynaptic sites of release. Other studies using super resolution microscopy found that trans-synaptic 'nanocolumns' aligned postsynaptic receptors and scaffolding proteins with presynaptic release sites (*Tang et al., 2016*), suggesting that certain proteins align with and may facilitate evoked release at spatially segregated sites. These data further support the differential organization of different modes of neurotransmitter release. Nevertheless, subsynaptic organization of different modes of release remain poorly understood as the existing tools for examining this fundamental property in real-time remain limited.

Here, we used a glutamate sensing fluorescent reporter, iGluSnFR (*Marvin et al., 2013*), to probe the spatial segregation of excitatory neurotransmission in hippocampal neurons. iGluSnFR is a novel probe that can resolve rapid glutamatergic transients with a high signal-to-noise ratio and is commonly used for in vivo studies (*Helassa et al., 2018*; *Marvin et al., 2018*). Using photobleaching of iGluSnFR as a tool to probe the organization of spontaneous and evoked release sites, we test whether fluorescent iGluSnFR probes within subsynaptic regions are differentially affected by photobleaching. We demonstrate that photobleaching has use-dependent properties, that iGluSnFR is a highly mobile probe expressed at synaptic surfaces, and that iGluSnFR demonstrates single synapse level resolution. Furthermore, iGluSnFR imaging can reveal the organization of distinct forms of neurotransmission at the synapse, creating a broader framework to elucidate basic principles of neurotransmission.

## Results

### iGluSnFR localization at the plasma membrane

Primary hippocampal neuron cultures were sparsely transfected with the glutamate sensor iGluSnFR under control of a neuronal promoter (synapsin-1). Sparse labeling of neurons allows the localization of singular and nonoverlapping pre- and postsynaptic specifications (*Figure 1A*).

Using confocal imaging of fixed neuronal cultures, we examined whether the expression of iGluSnFR in transfected neurons affects synapse number. We compared synapse count in iGluSnFR transfected neurons, with two control groups: GFP transfected neurons, and nontransfected neurons immunostained for a dendritic marker (MAP2). We then immunostained for pre- and postsynaptic markers (VGluT1 and PDS95, respectively), and we measured the number of colocalized pre- and postsynaptic terminals per length of neurite to estimate synapse density (*Alten et al., 2021*). We found that iGluSnFR expression did not significantly change the density of excitatory synapses (*Figure 1—figure supplement 1A, B*), indicating that expression of iGluSnFR at the plasma membrane does not disrupt synaptic organization.

We next examined the resting spatial distribution of iGluSnFR probes at the plasma membrane of neurons using super resolution microscopy, which can resolve the localization of molecules past the diffraction limit of light (*Huang et al., 2010*). To visualize iGluSnFR via super resolution, we stained for the GFP protein in the iGluSnFR molecule. This allowed for quantification of all iGluSnFR molecules, both fluorescent (i.e., activated by glutamate) and not. After applying clustering analysis using density-based spatial clustering of applications with noise (DBSCAN) (*Sawant, 2014*), we observed that iGluSnFR has a greater density at synaptic regions (*Figure 1B, C*). Pair correlation analysis of iGluSnFR clusters that colocalized with the postsynaptic marker PSD95 (*Figure 1—figure supplement*

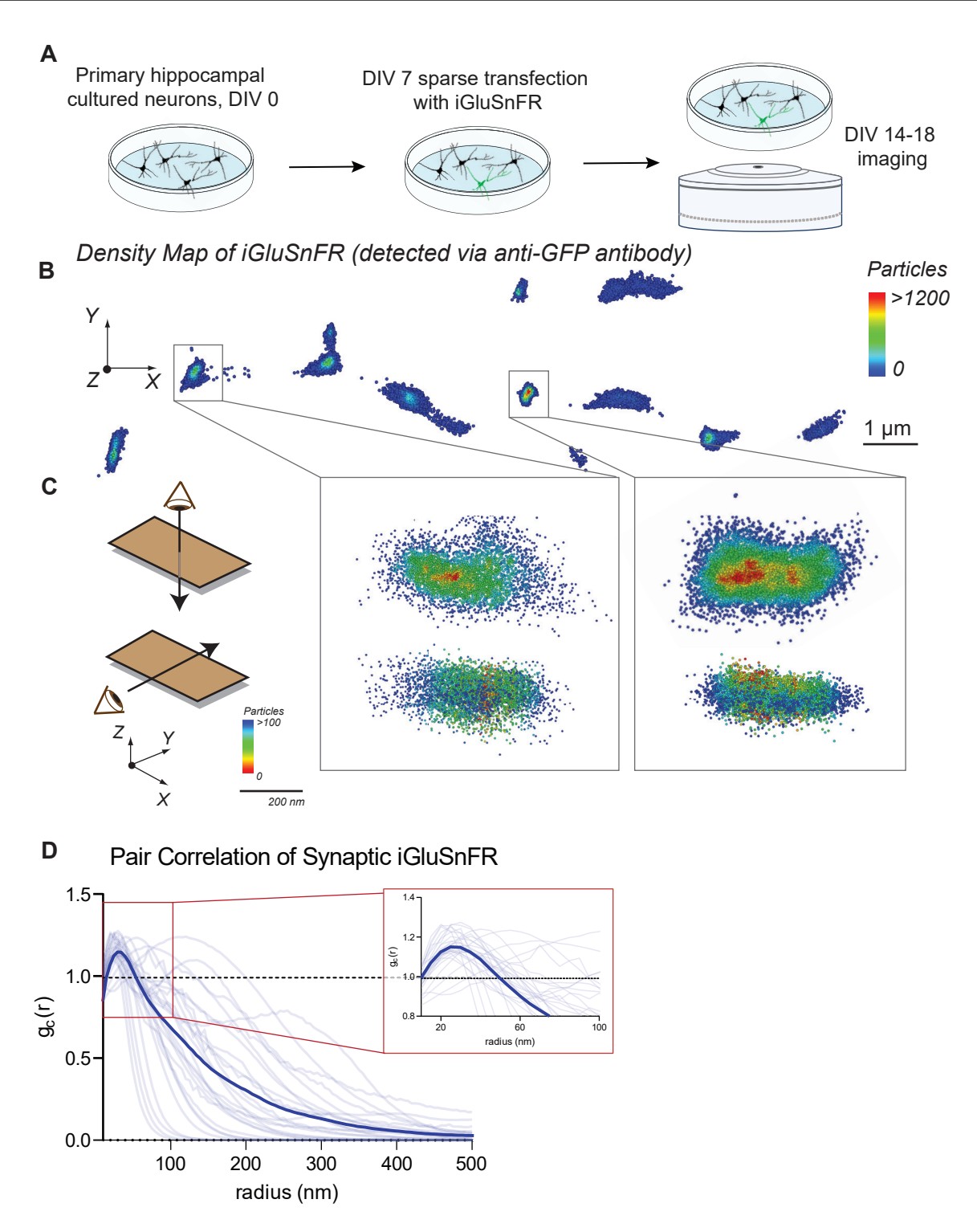

**Figure 1.** Distribution of iGluSnFR at the neuronal membrane. (**A**) Experimental paradigm and timeline for sparse transfection and subsequent imaging of iGluSnFR. (**B**) Representative image showing the density distribution of a density analysis of iGluSnFR probes at the postsynaptic membrane of an iGluSnFR transfected neuron, where areas in the red gradient have a denser distribution of probes and blue colored regions represents less dense regions. (**C**) Representative image of two synaptic boutons that colocalize to PSD95, showing that there is increased density of iGluSnFR probes closer to the center of the cluster. (**D**) Pair correlation of iGluSnFR clusters that colocalize to PSD95, demonstrating increasing clustering near the center of the synapse ($n$ = 30 synapses).

*Figure 1 continued on next page*

Figure 1 continued

The online version of this article includes the following source data and figure supplement(s) for figure 1:

**Source data 1.** Source data for *Figure 1*.

**Figure supplement 1.** Synapse density is unchanged as a result of iGluSnFR transfection.

**Figure supplement 1—source data 1.** Source data for *Figure 1—figure supplement 1*.

**Figure supplement 2.** iGluSnFR co-localizes with a synapse marker and demonstrates clustering.

*2A, B*), revealed an increased clustering of iGluSnFR near the center of the synapse, compared to a theoretical random distribution (*Figure 1D*). This suggests the existence of a general trapping mechanism at dendritic spines that confines iGluSnFR.

## iGluSnFR can resolve evoked and spontaneous events at the single synapse level in hippocampal neurons

Live imaging of primary neurons sparsely transfected with iGluSnFR was performed using an epifluorescence microscope equipped with an EM-CCD camera; this reduces photobleaching and toxicity during long experiments and allows resolution of fast spiking events, respectively. For this work, we focused on individual dendritic spines to isolate glutamatergic events postsynaptically. To find active synapses, we delivered a high-frequency stimulation (20 Hz, 25APs) followed by 90 mM KCl at the end of each recording, and used it to draw circular regions of interests (ROIs) of 2 μm diameter around local fluorescence maxima in dendritic spines. We then used these ROIs to analyze evoked and spontaneous activity over time (*Figure 2A*). To examine whether iGluSnFR can resolve evoked transmission at single synapses, we estimated release probability (Pr), or the likelihood that a docked and primed synaptic vesicle will fuse with the plasma membrane upon depolarization from an action potential. Pr was estimated using failure analysis, where 20–40 APs at 0.2 Hz were delivered and the number of detected responses was divided by the number of stimulations. Peaks, corresponding to glutamate release events, were detected with a high signal-to-noise ratio, and peaks were included at amplitude values ≥3 standard deviations above the baseline (*Figure 2B–D*). Pr has been widely reported to be low and highly variable at individual hippocampal synapses, in culture as well as in intact tissue, ranging from 0.1 to 0.2 to 0.6 (*Branco et al., 2008*; *Chanaday and Kavalali, 2018*; *Farsi et al., 2021*; *Jensen et al., 2021*; *Leitz and Kavalali, 2011*; *Murthy et al., 1997*; *Tagliatti et al., 2020*). Accordingly, iGluSnFR reported a low average release probability that was also highly variable among synapses (*Figure 2E–G*), indicating that each ROI contained a single synapse.

To extend previous studies and validate synaptic transmission measurements using iGluSnFR, we systematically determined whether we could modulate the detected events. Pr is positively regulated by extracellular $Ca^{2+}$ levels (*Chanaday and Kavalali, 2018*; *Leitz and Kavalali, 2011*; *Murthy et al., 1997*). In iGluSnFR transfected neurons, perfusion of solutions with increasing $Ca^{2+}$ concentrations leads to a corresponding increase of Pr (*Figure 2E, F*). Glutamate release is completely abrogated in the absence of extracellular calcium (*Figure 2F*), while Pr values are shifted toward high probabilities of 0.86 ± 0.23 in 8 mM $Ca^{2+}$, compared to 2 mM at 0.49 ± 0.32 (*Figure 2G, H*). We also measured the amplitude of evoked events, which correlates to the amount of glutamate released at synapses. We found that the peak amplitudes increased at higher extracellular $Ca^{2+}$ concentrations (*Figure 2I*). Based on the literature, this could be due to multivesicular release as release probabilities increase at higher $Ca^{2+}$ concentrations (*Leitz and Kavalali, 2011*; *Rudolph et al., 2015*), or glutamate spill over from adjacent sites (*Armbruster et al., 2020*), or a combination of these factors.

Next, we examined spontaneous neurotransmission detected by iGluSnFR. We were able to detect individual spontaneous glutamate release events at synapses with a high signal-to-noise ratio, similar to that of evoked events (*Figure 3A, B*). The average frequency of spontaneous events has been reported to be approximately 0.01–0.02 Hz per synapse (*Geppert et al., 1994*; *Leitz and Kavalali, 2014*; *Murthy and Stevens, 1999*; *Reese and Kavalali, 2016*) and our detection of spontaneous events using iGluSnFR is comparable to this value (*Figure 3C*). Due to the low probability of release, responses to action potentials at the level of individual synapses is a binary process: release of a single synaptic vesicle (one quanta) or a failure, as discussed above. At this scale, responses are fundamentally different from whole-cell measurements, where integrated signals from hundreds of synapses are measured. Thus, at individual synapses, both evoked and spontaneous neurotransmission lead

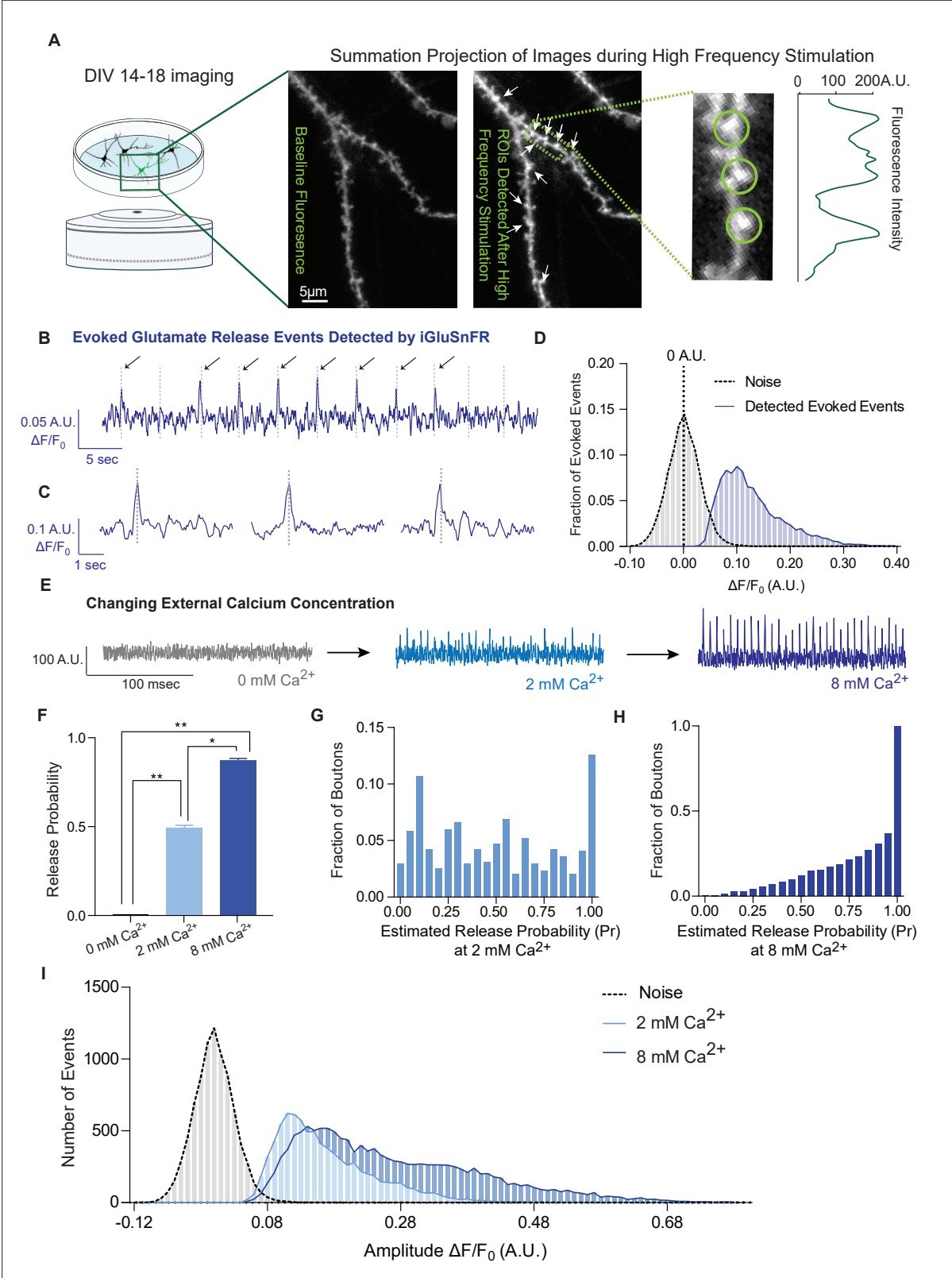

**Figure 2.** iGluSnFR detects excitatory evoked events with single synapse level resolution. (**A**) Summation projection of acquired images during high-frequency stimulation to locate active synaptic boutons, as detected as local fluorescence maxima. (**B**) Representative evoked traces of iGluSnFR, demonstrating the presence of failures and successes in response to stimulation (time of stimulation is marked by dashed lines). (**C**) Individual evoked traces to represent the kinetics of detected glutamatergic events. (**D**) Amplitudes of detected evoked events are readily distinguishable from noise

*Figure 2 continued on next page*

*Figure 2 continued*

of traces from single synapse recordings (*n* = 10 coverslips). (**E**) Representative single synapse recordings in the presence of changing external Ca$^{2+}$ concentrations, as both the number and size of events increase with increasing Ca$^{2+}$. (**F**) Release probability increases with increasing Ca$^{2+}$ (*n* = 9 coverslips). (**G**) Distribution of estimated release probability across single synapses in 2 mM Ca$^{2+}$ (*n* = 9 coverslips with a total of 636 synapses). (**H**) Distribution of estimated release probability across single synapses in 8 mM Ca$^{2+}$ (*n* = 9 coverslips with a total of 636 synapses). (**I**) Histogram of event sizes in the presence of increasing Ca$^{2+}$, as well as the noise of the traces (*n* = 9 coverslips) Bar graphs are mean ± standard error of the mean (SEM). Significance levels were stated as follows: *$p < 0.05$, **$p < 0.01$. ns denotes nonsignificance.

The online version of this article includes the following source data for figure 2:

**Source data 1.** Source data for *Figure 1*.

to release of one synaptic vesicle, thereby activating a similar numbers of probes and leading to similar event amplitudes (*Leitz and Kavalali, 2011*). Based on this, we expect to find similar results using iGluSnFR, and indeed we measured comparable peak amplitudes for evoked and spontaneous events (*Figure 3D*). This provides further support that we are detecting the release of single synaptic vesicles at single synapses. When release properties were analyzed at the level of individual synapses, spontaneous neurotransmission rate and evoked release probability did not correlate within synapses (*Figure 3E*), similar to prior studies (*Leitz and Kavalali, 2014*; *Reese and Kavalali, 2016*), thus reinforcing that evoked and spontaneous neurotransmission are partially segregated. Importantly, we observed that the spontaneous event frequency in a bath solution containing CNQX and APV (which blocks AMPAR and NMDARs, respectively) is not significantly different than in CNQX, APV, and the action potential blocker tetrodotoxin (TTX (*Figure 3F*)), demonstrating that inhibition of ionotropic glutamate receptors is sufficient to suppress action potential-driven release. Furthermore, these drugs do not interact with iGluSnFR (*Marvin et al., 2013*). This allowed us to monitor spontaneous neurotransmission at the same neurons and synapses used for studying evoked neurotransmission, since TTX is hard to wash off. Thus, all studies on spontaneous neurotransmission using iGluSnFR were performed in the presence of only CNQX and APV.

To further evaluate the authenticity of spontaneous, action potential-independent release events detection using iGluSnFR, we next assessed the modulation of spontaneous neurotransmission by reducing extracellular Ca$^{2+}$ concentration or perfusing hypertonic sucrose (*Figure 3G*). In 0 mM Ca$^{2+}$, spontaneous release propensity is lower but not completely diminished, and iGluSnFR event amplitudes remained unchanged (*Figure 3H, I*). This is consistent with previous studies demonstrating that spontaneous release is less sensitive to reductions in bath Ca$^{2+}$ than evoked release (*Kavalali, 2020*; *Xu et al., 2009*). We also perfused cells with a hypertonic sucrose (+100 mOsm) solution after recording baseline activity. Hypertonic sucrose has been shown to trigger the fusion of docked synaptic vesicles in an action potential and Ca$^{2+}$-independent manner (*Rosenmund and Stevens, 1996*). While high hypertonicity (+300 to +500 mOsm) causes the whole readily releasable pool to rapidly fuse, individual release events can be resolved at lower sucrose concentrations (+100 mOsm) (*Rosenmund and Stevens, 1996*). Accordingly, upon perfusion of +100 mOsm hypertonic sucrose, the spontaneous event frequency detected by iGluSnFR increased, while the amplitude of detected events remained the same (*Figure 3J, K*).

These results expand upon earlier work that validated iGluSnFR at single synapses (*Farsi et al., 2021*; *Tagliatti et al., 2020*), and demonstrate that iGluSnFR detects quantal release for both evoked and spontaneous neurotransmission. In these experiments, we also observed that the detection of these different modes of release can be modulated by established parameters, such as Ca$^{2+}$ and hypertonicity. These results show that iGluSnFR is a reliable and useful tool in measuring neurotransmitter release from single synapses.

## iGluSnFR is a highly mobile probe in the plasma membrane but demonstrates an immobile fraction at synapses

When excited by light, fluorophores can be photobleached, removing them from the normal emission cycle (*McQuarrie and Simon, 1997*). In contrast, fluorophores in the ground state (or nonfluorescent) cannot be photobleached, meaning that by its very nature, photobleaching is a use-dependent process (*Figure 4A*). To measure the mobility of iGluSnFR at the plasma membrane, we employed fluorescence recovery after photobleaching (FRAP), which can measure fluorophore diffusion by examining the rate at which fluorescent probes replace photobleached probes within a selected region

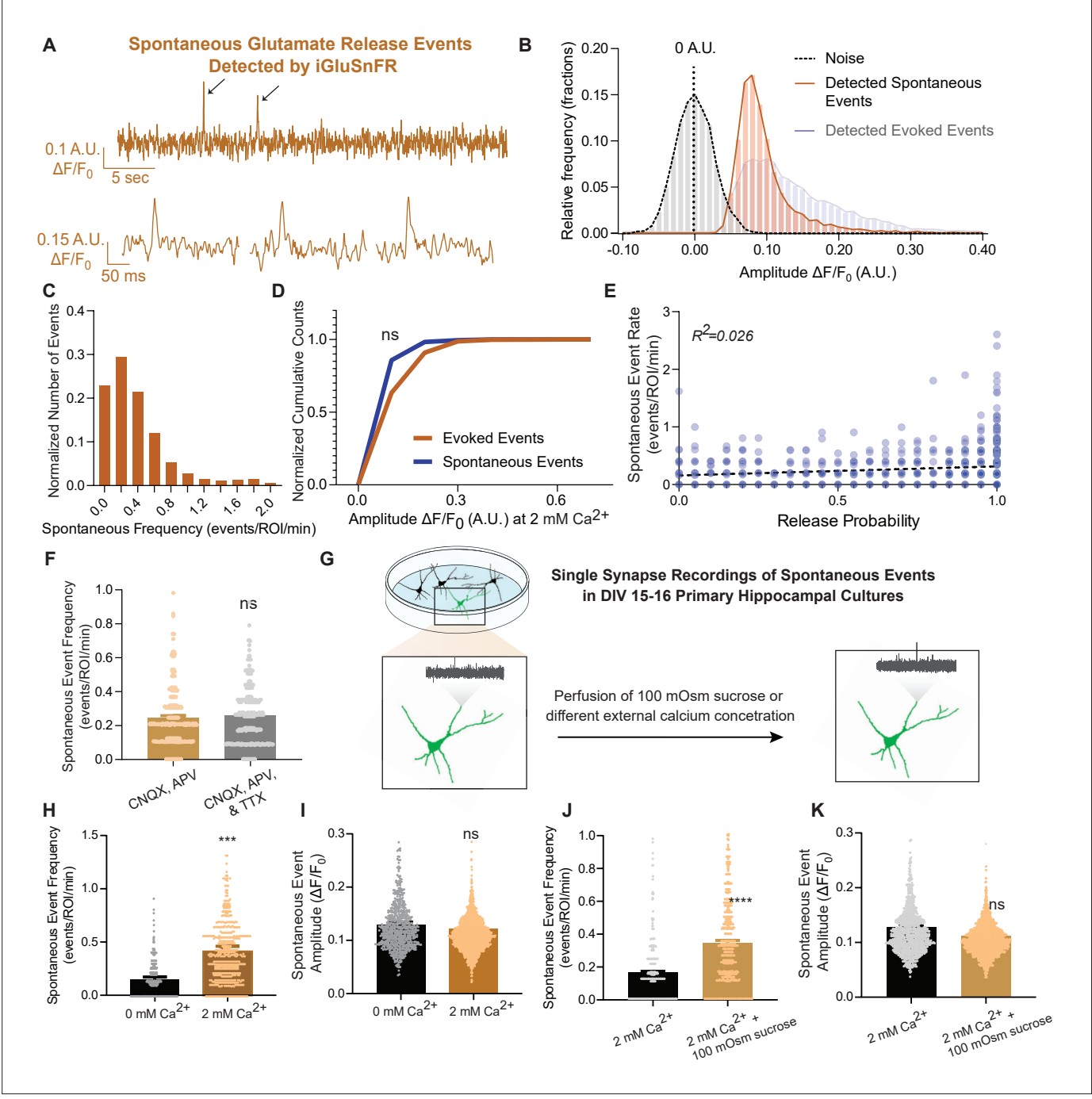

**Figure 3.** iGluSnFR can resolve spontaneous events at single synapses. (**A**) Representative spontaneous trace, and representative individual spontaneous events. (**B**) Amplitudes of detected spontaneous events are readily distinguishable from the noise of the trace (*n* = 10 coverslips). (**C**) Distribution of spontaneous rate at single synapses across multiple recordings. (**D**) Cumulative histogram of spontaneous and evoked event size compared to each other (*n* = 19 coverslips per group, KS of coverslip averages: p = 0.9). (**E**) Within synapses, the spontaneous event rate and estimated release probability demonstrate no linear correlation. (**F**) Comparison of spontaneous frequency between cultures in CNQX and APV, to the spontaneous frequency of those same cultures after perfusion of CNQX, APV, and TTX (*n* = 4 coverslips, Welch's *t*-test of coverslip averages: p = 0.4). (**G**) Experimental paradigm of validating spontaneous events with high sucrose and changing $Ca^{2+}$ concentrations. (**H**) The spontaneous rate can be decreased with a lower $Ca^{2+}$ concentration, compared to physiological concentrations at the same synapse over time. Individual points represent measurements from individual synapses (*n* = 6 coverslips, p = 0.009). (**I**) The spontaneous event size at changing $Ca^{2+}$ concentrations remain the same (*n* = 6 coverslips, p = 0.17). (**J**) Spontaneous event frequency after perfusing Tyrode's with 100 mOsm sucrose is increased (*n* = 8, p = 0.005). (**K**)

*Figure 3 continued on next page*

*Figure 3 continued*

Spontaneous amplitude after perfusing Tyrode's with 100 mOsm is not significantly different (*n* = 8, p = 0.18). Bar graphs are mean ± standard error of the mean (SEM). Significance levels were stated as follows: ***p < 0.001, and ****p < 0.0001. ns denotes nonsignificance.

The online version of this article includes the following source data for figure 3:

**Source data 1.** Source data for *Figure 1*.

(*Axelrod et al., 1976*). Analysis was performed using a previously validated web-based software (*Koulouras et al., 2018*). The rate of recovery, which is a proxy for probe mobility, was estimated by fitting the curve with a single exponential equation, and the immobile fraction of probe was defined as the difference between baseline (prebleaching) and the asymptotic (postbleaching) normalized fluorescence (*Figure 4B*). Using neurons transfected with iGluSnFR, we selected neuronal dendritic spines and shaft areas to bleach (ROI size = 1 µm diameter for FRAP experiments only) and monitored fluorescence recovery over time using a confocal microscope (*Figure 4C, D*). In neurons, iGluSnFR is a very mobile probe, with similar recovery time constants between spine and shaft regions averaging 8.9 ± 7.0 and 9.9 ± 6.1 s, respectively (*Figure 4E*). iGluSnFR also exhibited an appreciable immobile fraction of 23% and 24% in spinous and shaft regions, respectively (*Figure 4F*). Immobile fractions reflect a subpopulation of probes that cannot be replenished by nonbleached probes. The existence of this immobile set of fluorophore molecules could be due to the geometry of the neuronal structure, for instance due to limitation of diffusion by surface proteins (*Chen et al., 2021*; *Tardin et al., 2003*). Another explanation could be that this is an intrinsic property of the probe, in which it interacts with proteins that tether it to the membrane surface.

To gain insight into the source of the iGluSnFR immobile fraction, we measured the mobility of the probe in a different cell system (*Figure 4G, H*). We found that iGluSnFR is slightly more mobile in HEK cells (time constant of 5.6 ± 3.6 s) than in neurons (*Figure 4H, I*). This suggests that iGluSnFR likely does not have an intrinsic property that leads it to tether to intracellular proteins, and that limitations in diffusion may be partially due to the geometry and asymmetric composition of neuronal membranes. In favor of this speculation, using stochastic optical reconstruction microscopy (STORM) we observed the iGluSnFR was concentrated at synapses, pointing to a putative mechanical trapping mechanism (*Figure 1*). We also observed that the immobile fraction was not significantly different in HEK cells compared to neurons (*Figure 4J*). Thus, the increased mobility of iGluSnFR in HEK cells compared to neurons suggests that it is less likely that there is an intrinsic property of the probe that tethers it to a specific location on the plasma membrane surface but it rather arises from neuron-specific diffusion barriers.

## Photobleaching can be used as a use-dependent tool to investigate the subsynaptic spatial segregation of evoked and spontaneous neurotransmission

Our results showed that iGluSnFR has single synapse level resolution, and that the probe is highly mobile within the plasma membrane with a significant immobile fraction. Based on this information, we next used photobleaching and its use-dependent properties to probe putative segregation of evoked and spontaneous neurotransmission. To do this, we measured spontaneous and evoked events in iGluSnFR transfected neurons using an epifluorescence microscope, with the same parameters used to validate single synapse neurotransmission. Synaptic ROIs were selected during high-frequency stimulation at the end of each experiment, and neuronal activity was measured retroactively. We recorded spontaneous neurotransmission for 10 min and subsequently monitored evoked release (*Figure 5A*) to minimize confounding effects on spontaneous release by the electrical stimulation. We then photobleached the entire field of view with sustained maximal-intensity illumination (by removing the light dimming filter), and then resumed normal imaging of spontaneous and evoked events from all the photobleached synapses (*Figure 5A*). This approach has the advantage of allowing the study of tens to hundreds of synapses, providing information about intrinsic biological variabilities. After 30 s of photobleaching, the detection of spontaneous events decreased significantly but not completely. At longer photobleaching periods of up to 20 min, spontaneous event frequency and amplitudes were decreased, although events remained detectable (*Figure 5B, C*). Evoked release probability decreased significantly after 30 s. And within 5–10 min, evoked release became largely undetectable

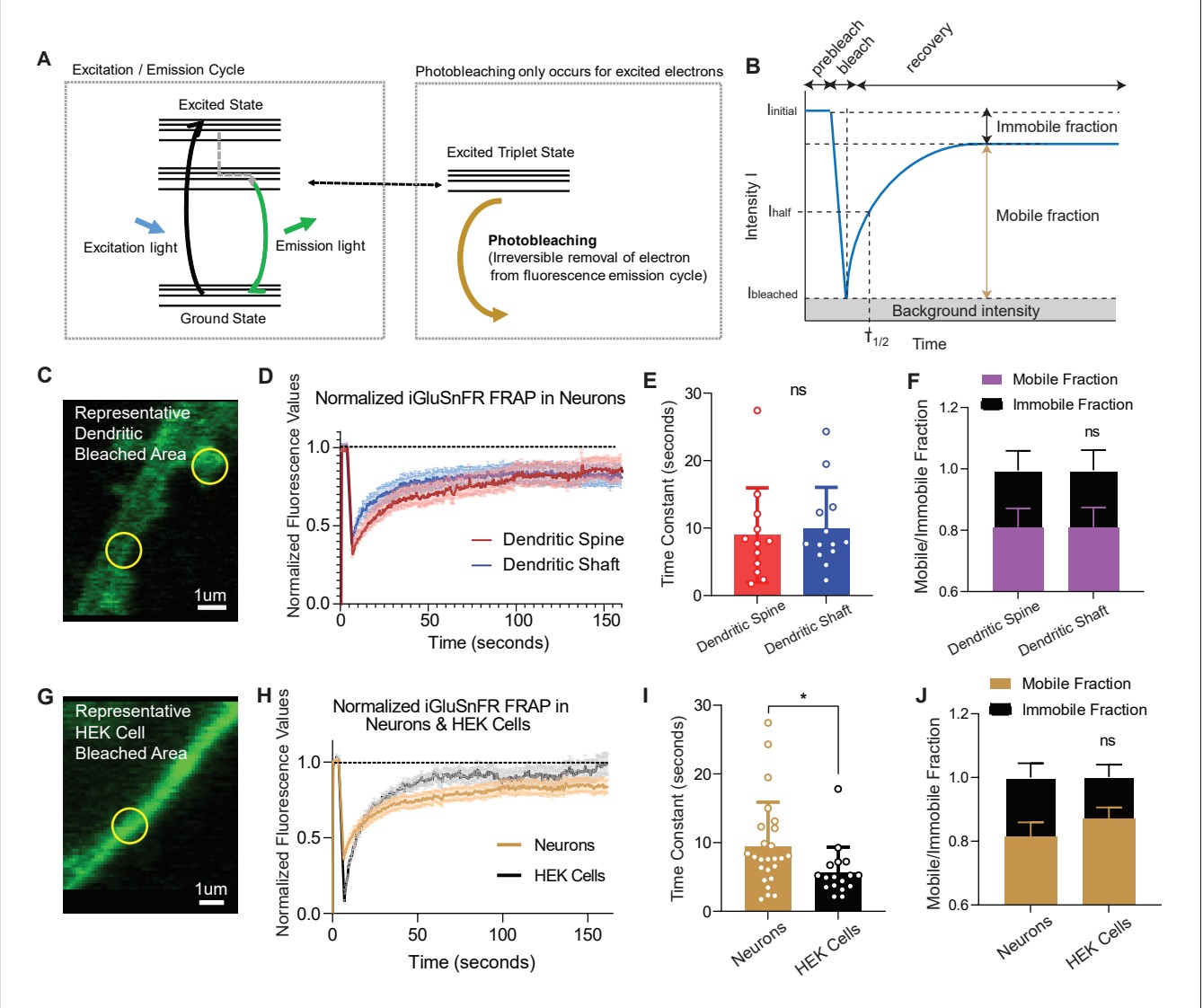

**Figure 4.** Fluorescence recovery after photobleaching (FRAP) experiments reveal iGluSnFR to be a highly mobile probe, and that there is an immobile fraction of this probe at neuronal synapses. (**A**) Photobleaching mechanism diagram. (**B**) Diagram illustrating analysis of the prebleaching and recovery curve of FRAP experiments, including how measurements of immobile/mobile fractions as well as the time constants are taken. (**C**) Representative bleached areas of neuron dendritic areas, both spinous and nonspinous regions. (**D**) Recovery curve of measured regions after photobleaching of iGluSnFR in neurons via FRAP, both spinous and nonspinous (shaft) regions. (**E**) Time constants of FRAP recovery in neurons transfected with iGluSnFR (mean ± standard deviation [SD], spinous regions $n = 12$: 8.9 ± 7.0, nonspinous regions $n = 13$: 9.9 ± 6.1; Welch's $t$-test p = 0.7). (**F**) Immobile fractions of bleached regions in neurons (mean ± SD, spinous regions $n = 12$: 0.19 ± 0.23, nonspinous regions $n = 13$: 0.19 ± 0.25; Welch's $t$-test p = 0.9). (**G**) Representative bleached region of HEK cells. (**H**) Recovery curve of bleached iGluSnFR in neurons compared to HEK cells. (**I**) Time constants of FRAP recovery in neurons and HEK cells transfected with iGluSnFR (mean ± SD; neuronal regions $n = 25$: 9.4 ± 6.4 s, HEK cell regions n = 17: 5.7 ± 3.6 s; Welch's $t$-test p = 0.02). (**J**) Immobile fractions between neurons and HEK cells (mean ± SD; neuronal regions $n = 17$: 0.19 ± 0.23, HEK cell regions regions $n = 25$: 0.13 ± 0.15; Welch's $t$-test p = 0.36). Significance levels were stated as follows: *p < 0.05. ns denotes nonsignificance.

The online version of this article includes the following source data for figure 4:

**Source data 1.** Source data for *Figure 4*.

(*Figure 5D, E*), demonstrating a significant difference in its rate of photobleaching compared to spontaneous neurotransmission (*Figure 5—figure supplement 1A*).

We next repeated the above set of experiments but while electrically stimulating during the photobleaching process (*Figure 5—figure supplement 1B*). We did not observe a difference in evoked release detection following photobleaching with stimulation versus without stimulation

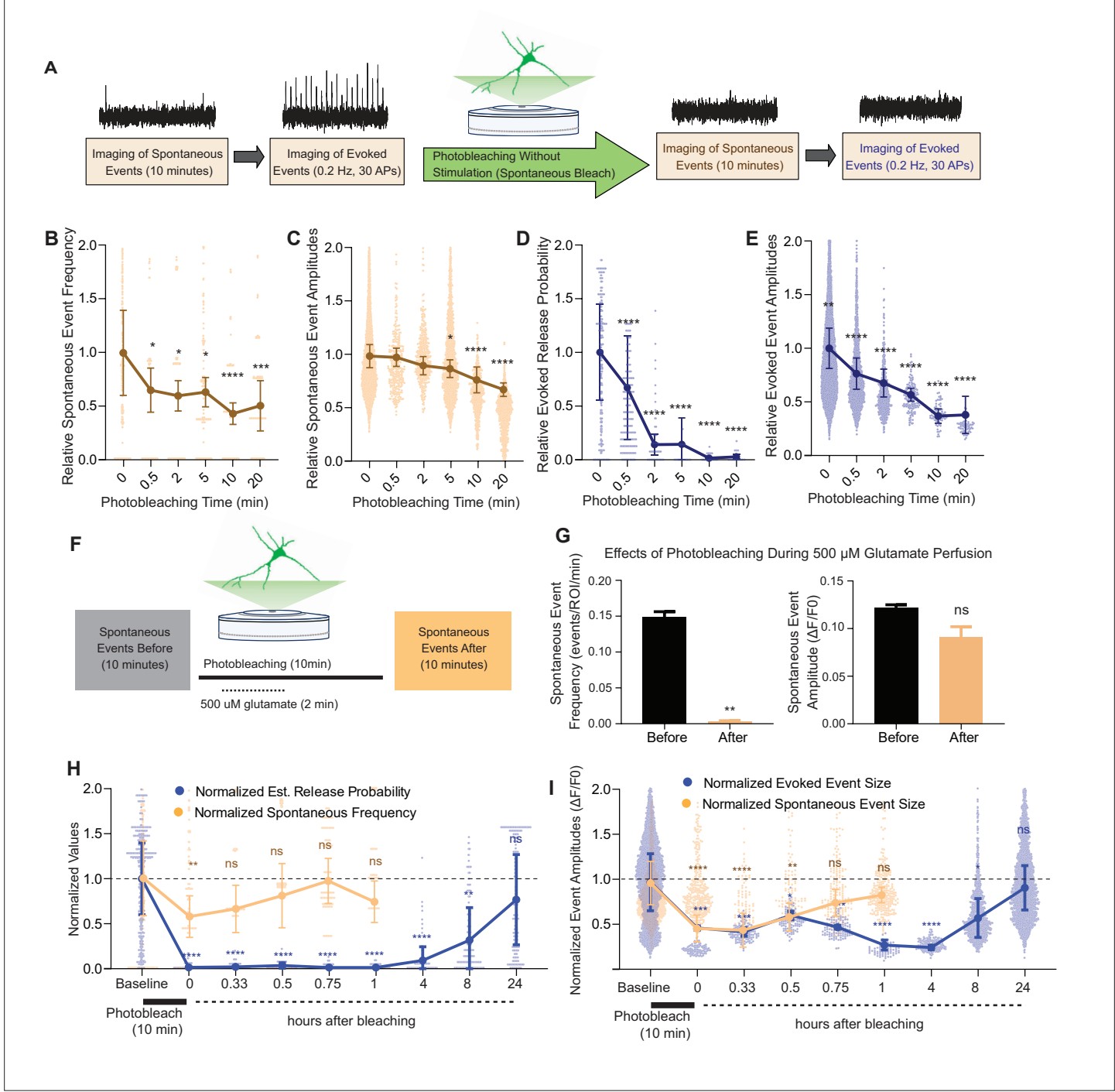

**Figure 5.** Spontaneous and evoked events are differentially bleached over time. (**A**) Experimental paradigm of photobleaching experiments. Spontaneous and evoked events were measured, followed by photobleaching, and then once again spontaneous and evoked events were measured. (**B**) Relative spontaneous event frequency after photobleaching of 0.5–20 min without stimulation. For *Figure 4B–I*, individual points represent measurements from individual synapses and statistics were done on averages of synapses in each coverslip. At least four coverslips were included for every group. (**C**) Relative spontaneous event sizes after photobleaching of 0.5–20 min without stimulation. (**D**) Relative release probability after photobleaching of 0.5–20 min without stimulation. (**E**) Relative event size amplitude after photobleaching of 0.5–20 min without stimulation. (**F**) Experimental paradigm of photobleaching while perfusing glutamate. (**G**) Spontaneous event frequency and event size while perfusing glutamate during photobleaching. (**H**) Recovery of release probability and spontaneous frequency, both normalized to values prior to photobleaching at the same synapse to account for differences in release across synapses. Spontaneous release recovers within minutes, while evoked release recovers within hours. (**I**) Recovery of evoked and spontaneous event sizes correlate with their release rates, with spontaneous frequency recovering within minutes and evoked

*Figure 5 continued on next page*

*Figure 5 continued*

release recovering within hours Bar graphs are mean ± standard error of the mean (SEM). Significance levels were stated as follows: **p < 0.05, **p < 0.01, ***p < 0.001, and ****p < 0.0001. ns denotes nonsignificance.

The online version of this article includes the following source data and figure supplement(s) for figure 5:

**Source data 1.** Source data for *Figure 5*.

**Figure supplement 1.** Comparison of spontaneous and evoked photobleaching on detected glutamate release.

**Figure supplement 1—source data 1.** Source data for *Figure 5—figure supplement 1*.

(*Figure 5—figure supplement 1G*). One explanation could be that spontaneous release during the photobleaching period is sufficient to occlude all evoked iGluSnFR responses. To test whether a higher baseline release probability could allow more discernment of the effect of stimulating or not during photobleaching, we repeated the experiments in a higher extracellular $Ca^{2+}$ of 8 mM. In this condition, probability of release is close to 1 during baseline recordings (*Figure 2H*). In agreement, we observed a higher degree of photobleaching of evoked responses when electrical stimulation was applied during the bleaching period in the presence of 8 mM extracellular $Ca^{2+}$ compared to nonstimulated neurons (*Figure 5—figure supplement 1I*), indicating that the level of photobleaching of evoked responses scales up with the amount of evoked glutamate release. These results could also be explained by reduced action potential firing due to reduced excitability in the presence of high calcium (*Segal, 2018*).

Our results so far indicate that spontaneous neurotransmission detected using iGluSnFR is more resilient to photobleaching than its evoked counterpart. One interpretation is that iGluSnFR molecules activated by spontaneous release are in more mobile areas of the membrane and thus can be rapidly replaced by unbleached probes from other regions. If this is the case, activating all iGluSnFR molecules during the photobleaching period should abolish future detection of spontaneous neurotransmission. To test this hypothesis, we recorded spontaneous events followed by a total of 10 min of photobleaching, during which for 2 min we perfused 500 μM glutamate (*Figure 5J*). We found that the detection of spontaneous iGluSnFR events was significantly and nearly completely diminished after photobleaching during glutamate perfusion (*Figure 5K*). Thus, activating all iGluSnFR probes at the plasma membrane during photobleaching eliminates event detection. Importantly, this finding serves as a key negative control validating the specificity of the photobleaching approach toward bona fide glutamate release events.

## Fluorescence at evoked and spontaneous release sites recover at different timescales after photobleaching

We next aimed to quantify the time course of recovery after photobleaching of spontaneous and evoked iGluSnFR events. For this purpose, we used a 10-min photobleaching period. Following photobleaching, we waited for 20 min and up to 24 hr in the absence of fluorescence imaging (i.e., in the dark), then monitored glutamatergic activity in the same neurons. Both spontaneous and evoked events were significantly bleached following the 10-min photobleaching period, although evoked events were relatively more susceptible to photobleaching confirming our previous findings. Within an hour, spontaneous events were detectable at a frequency and amplitude that was similar to that prior to photobleaching (*Figure 5H–I*). On the other hand, evoked events remained undetectable an hour after photobleaching. After 8 hr, detection of evoked events resumed. At 24 hr after photobleaching, evoked event values returned to their original pre-photobleaching values (*Figure 5H, I*). By plotting the event rate over time, we found that the recovery of spontaneous events occurred on the timescale of minutes, while the recovery of evoked events occurred on a longer timescale of hours. The slower recovery rate of evoked events suggests that iGluSnFR probes that respond to evoked release have less lateral mobility at the plasma membrane. This is consistent with our previous data demonstrating an immobile fraction of iGluSnFR, which may be in part due to the evoked release architecture at synapses.

In the next set of experiments, we examined whether photobleaching altered the structure and function of synapses using electrophysiological detection of glutamatergic neurotransmission. To do this, we photobleached neurons plated on gridded coverslips on the epifluorescence microscope and located the same region and neuron for electrophysiological measurements on a separate

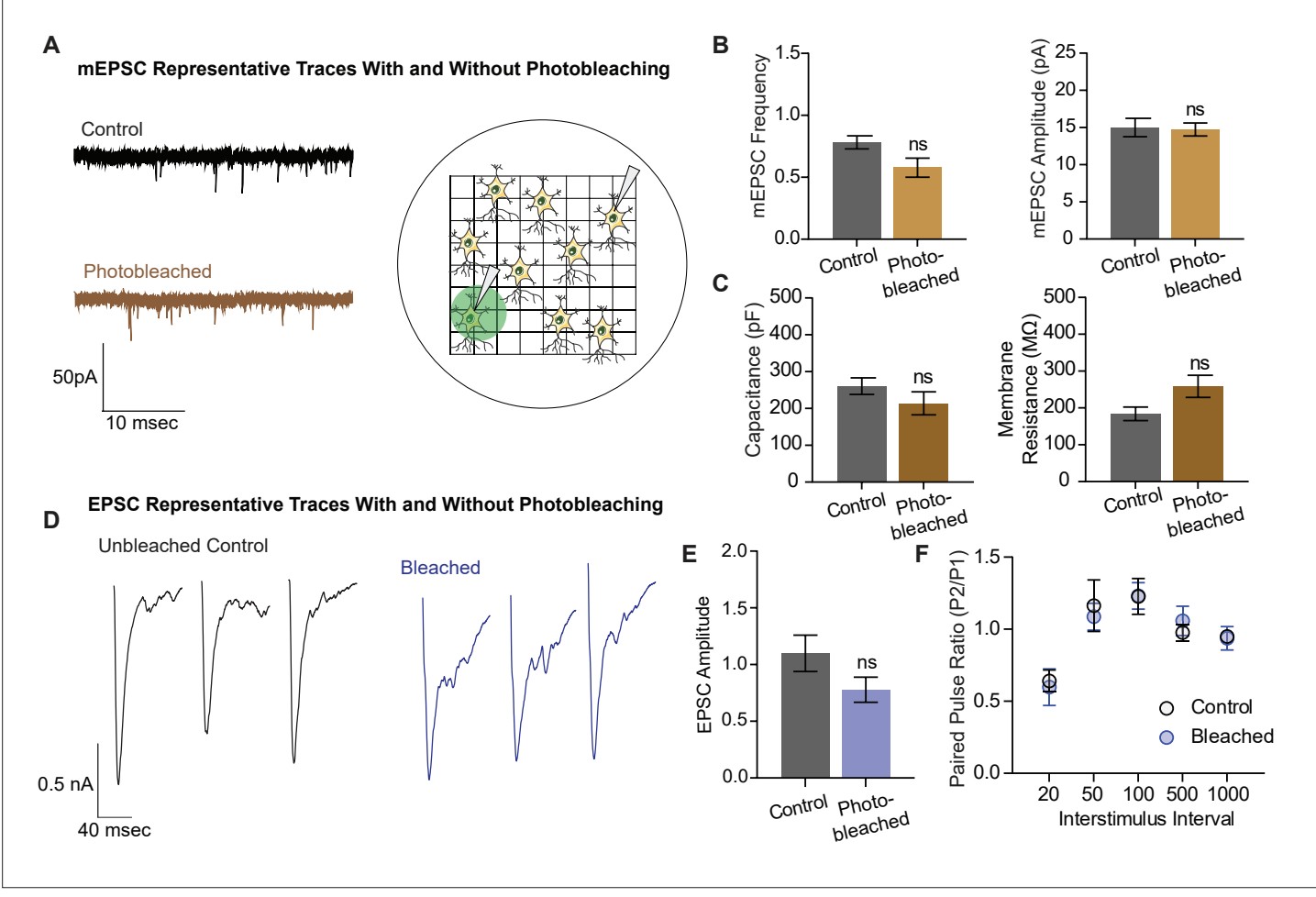

**Figure 6.** Electrophysiological and structural properties of neurons remain intact after photobleaching. (**A**) Representative miniature excitatory postsynaptic current (mEPSC) traces with and without photobleaching, and the experimental paradigm of recording electrophysiological measurements in photobleached versus nonphotobleached neurons. (**B**) mEPSC frequency and amplitude between unbleached and bleached neurons are not significantly different (control $n$ = 8, photobleached $n$ = 12; p = 0.066). (**C**) Capacitance and membrane resistance, markers of cell health, are not significantly different between unbleached and bleached neurons (capacitance $n$ = 11 for both groups, membrane resistance control $n$ = 10, photobleached $n$ = 11). (**D**) Representative EPSC traces with and without photobleaching. (**E**) EPSC amplitudes with and without photobleaching ($n$ = 9 for both groups). (**F**) Paired pulse ratios across different interstimulus intervals between neurons that have and have not been photobleached (control $n$ = 8, bleached $n$ = 9). Bar graphs are mean ± standard error of the mean (SEM). ns denotes nonsignificance.

The online version of this article includes the following source data for figure 6:

**Source data 1.** Source data for *Figure 6*.

electrophysiology rig (*Figure 6A*). We recorded miniature excitatory postsynaptic currents (mEPSCs), or spontaneous neurotransmission, in both photobleached and control neurons, and observed no difference in the frequency nor the amplitude of mEPSCs between the groups (*Figure 6B*). The capacitance and membrane resistance of neurons are indicative of membrane integrity and overall cell health, and these values were similar between control and photobleached neurons (*Figure 6C*). Finally, we examined evoked excitatory neurotransmission (*Figure 6D*), and found no difference in the amplitude of evoked postsynaptic currents, or EPSCs (*Figure 6E*). There was also no difference in the paired pulse ratio, which is a proxy for release probability, between the groups (*Figure 6F*). These data show that photobleaching does not affect excitatory synaptic transmission as measured by electrophysiology, nor the resting properties of neurons. Thus, the decreases seen in iGluSnFR event detection reflect the effects of photobleaching on fluorescent probes, and not intrinsic changes in neurotransmission levels due to toxic effects of prolonged illumination.

## Discussion

In this study, we show that iGluSnFR can be used as a tool to investigate the spatial segregation of spontaneous and evoked neurotransmission. We extend the validation of iGluSnFR's ability to resolve single synapses in hippocampal neurons (*Farsi et al., 2021*; *Tagliatti et al., 2020*), creating a custom MATLAB script that can reliably detect both evoked and spontaneous release. At individual synapses, spontaneous release occurs at very low frequency; thus, it is critical to have a robust negative control. We found that spontaneous events are decreased in lower $Ca^{2+}$ concentrations. We also show that we nearly eliminate the detection of spontaneous events by photobleaching while perfusing glutamate, thus activating (and thereby making available to photobleach) all probes on the neuronal surface. These results further support the specificity of our event detection, especially spontaneous events.

Using FRAP, we established that iGluSnFR is a highly mobile probe and can replenish bleached areas within seconds, and it also possesses a sizable immobile fraction in neurons. iGluSnFR is bound to the plasma membrane by a PDGFR domain, but it is not specifically targeted to any neuronal protein; it thus theoretically should be able to freely diffuse across the entirety of the neuronal surface. We took advantage of this property of iGluSnFR and measured evoked and spontaneous release detection in varying photobleaching conditions. Photobleaching is a use-dependent process, in which only actively fluorescent probes can be bleached, and nonfluorescent probes will not be affected. While photobleaching can be an artifact of imaging that we seek to minimize, it has been used intentionally in the past – for instance, to decrease background fluorescence (*Gandhi and Stevens, 2003*). We employed photobleaching as a use-dependent blocker of fluorescence detection to probe synaptic physiology. When we photobleached neurons transfected with iGluSnFR, we found a differential effect on evoked and spontaneous neurotransmission, in which evoked release events photobleach much faster and more efficiently than spontaneous release events (*Figure 4B–E*; note that iGluSnFR activated by tonic/ambient glutamate is also bleached – *Figure 5—figure supplement 1J* –). Furthermore, evoked release also takes longer to 'recover' its photobleached fluorescence when reimaged after a certain time period of time in the dark (*Figure 5A–B*). Previously, pharmacological use-dependent blockers and electrophysiological techniques have been used to uncover a functional segregation between these modes of neurotransmission (*Atasoy et al., 2008*; *Horvath et al., 2020*; *Peled et al., 2014*; *Reese and Kavalali, 2016*; *Sara et al., 2011*). By using optical techniques, we can obtain key information about spatial location of synaptic release, as well as indirectly probe the location of presynaptic glutamate release by examining glutamate responses at the postsynapse.

Our results build upon earlier work that show evoked release may be localized to specific and subsynaptic regions. For instance, *Tang et al., 2016* found that pre- and postsynaptic proteins align in nanocolumns that preferentially allow evoked release to occur. In agreement with this finding, a recent study identified LRRTM2 as a trans-synaptic adhesion protein that regulates AMPAR positioning at release sites (*Ramsey et al., 2021*). Several studies have demonstrated a mobile and immobile fraction of AMPARs (*Chen et al., 2021*; *Opazo and Choquet, 2011*), which may coincide with the immobile fraction of iGluSnFR found in our studies. Collectively, these studies suggest that certain proteins align to facilitate evoked release within spatially restricted regions.

Our findings also revealed a consistent subsynaptic structure in which evoked release is more clustered and restricted than spontaneous release, as such an organization would lead to photobleaching having a stronger effect on evoked release. Photobleached probes within the evoked release site have greater difficulty diffusing out of the bleached region and allowing unbleached probes to enter. Thus, within a few minutes, evoked release reaches a nearly undetectable level, and it takes a much longer time for the signal to recover. In contrast, spontaneous release may occur more broadly across the synapse, likely with less pronounced surface protein alignment at the site of release (*Figure 7*). These events are harder to photobleach because the mobility of iGluSnFR is high, such that by the time another spontaneous event occurs, unbleached iGluSnFR will have replenished the photobleached probes and be able to respond to subsequent spontaneous release events. iGluSnFR moves faster, on the order of seconds, than the rate of photobleaching which occurs on the order of minutes. Another explanation could be that there is a greater number of spontaneous release sites compared to evoked release sites, so that even if one release site is bleached, there are many others that have not been affected and could still fluoresce upon glutamate release. However, the latter proposal is inconsistent with the observation that 10–15 min long application of the use-dependent N-methyl-D-aspartate receptor (NMDA) receptor blocker MK-801 or folimycin, a use-dependent blocker of

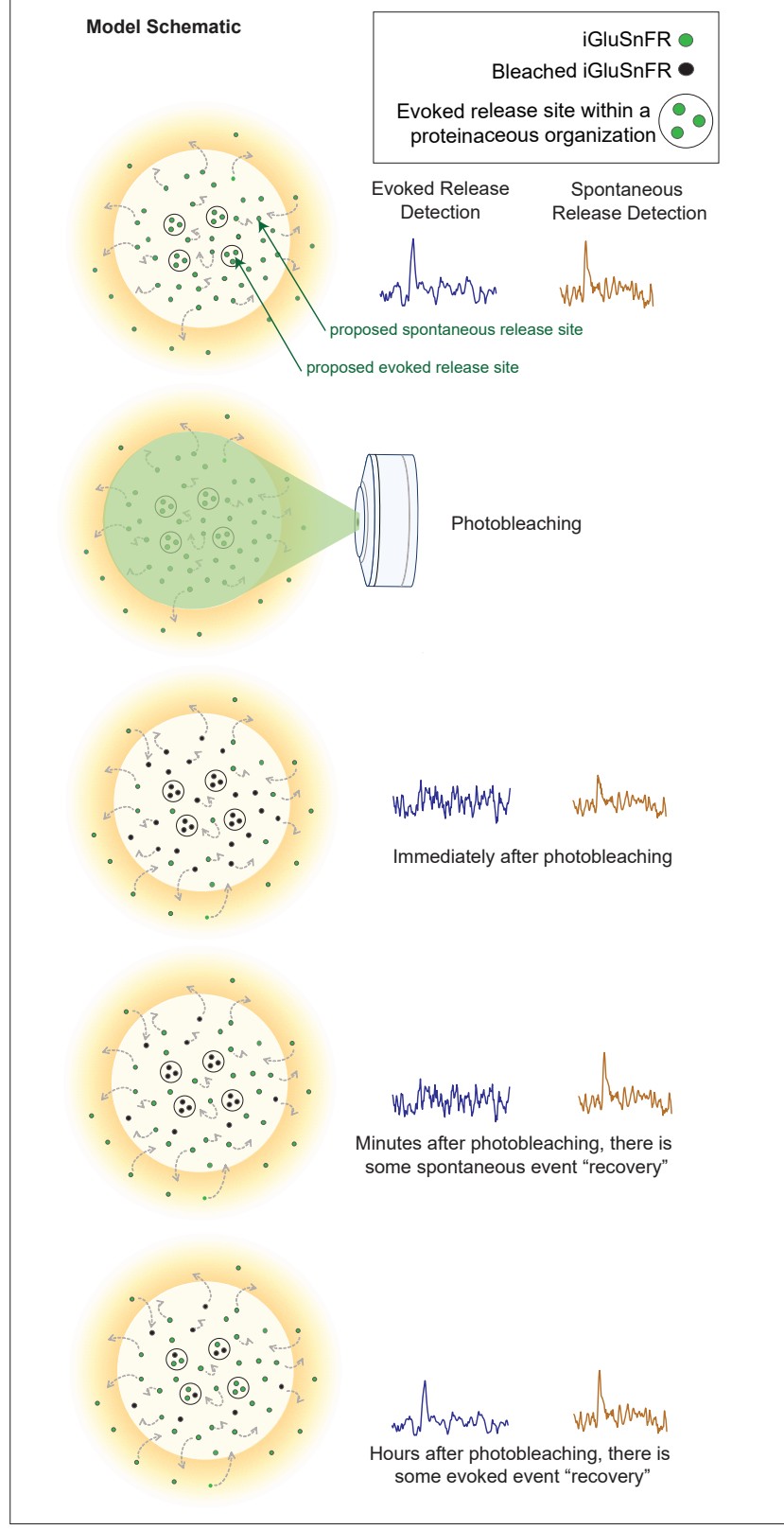

**Figure 7.** Model schematic demonstrating the clustered and diffusion restricted organization of evoked events, compared to the more diffusely located and freely moving structure of spontaneous event sites. Evoked events are more readily photobleached, likely due to its clustered location and less diffusible surface. Spontaneous events are still detectable after photobleaching, likely due to the more freely diffusible nature of its structure and their

*Figure 7 continued on next page*

*Figure 7 continued*

dispersed location. Recovery for spontaneous release occurs on the order of minutes, likely due to synaptic and extrasynaptic diffusion of unbleached probes into the bleached regions. Recovery for evoked release occurs on the order of hours.

presynaptic vesicle acidification, are sufficient to suppress spontaneous neurotransmission (*Atasoy et al., 2008*; *Sara et al., 2005*). Therefore, our photobleaching period of 20 min is beyond this time period and should have been sufficient to suppress all events if photobleached iGluSnFRs were not rapidly replenished. Finally, the time for evoked iGluSnFR events to recover fluorescence takes longer than for spontaneous events, which further suggests that there is a greater diffusion barrier within evoked release sites compared to spontaneous sites.

Our experiments were performed in an in vitro culture system as much prior synaptic work has been done in primary cultures, thus our results more directly build upon previous work (*Ramsey et al., 2021*; *Renner et al., 2017*; *Tang et al., 2016*; *Zhang et al., 2018*). While we would expect the findings from primary cultured neurons to largely translate to an intact brain system, there may be some differences. For instance, the differences in extracellular matrix composition may affect diffusional properties, and the 3D nature of synapse organization within the dense neuropil could lead to differences in glutamate dynamics compared to a monolayer of neurons (*Matthews et al., 2022*). It is also possible that more complex ex vivo brain tissues may introduce technical complications in detecting differences in photobleaching and fluorescence recovery of iGluSnFR probes. Furthermore, imaging synapses in intact brain systems would be better performed on a two-photon microscope, and this would require a recharacterization of iGluSnFR dynamics and photobleaching properties by a different microscope.

The differences in event detection after photobleaching and fluorescence recovery, as well as a clustered distribution uncovered by super resolution analysis, support a model of synaptic organization where different modes of neurotransmission are segregated. Here, we demonstrate that photobleaching can be used to probe the spatial segregation of different modes of neurotransmission. These results demonstrate a novel use for a widely recognized property of fluorescence probes, expanding the ways in which we use photobleaching as a molecular tool, as well as demonstrating the spatial organization of different modes of neurotransmission.

## Materials and methods

**Key resources table**

| Reagent type (species) or resource | Designation | Source or reference | Identifiers | Additional information |
|---|---|---|---|---|
| Antibody | Anti-GFP (rabbit polyclonal) | Synaptic Systems | Catalog 132 002 | ICC (1:300) |
| Antibody | Anti-PSD95 (mouse monoclonal) | Synaptic Systems | Catalog 124 011 | ICC/IHC (1:100) |
| Antibody | MAP2 (guinea pig polyclonal) | Synaptic Systems | Catalog 188 004 | IHC (1:500) |
| Antibody | vGluT1 (rabbit polyclonal) | Synaptic Systems | Catalog 135 302 | IHC (1:500) |
| Antibody | AlexaFluor 647 donkey anti-rabbit IgG (H + L) (polyclonal, secondary antibody) | Invitrogen | REF A31573 | IHC/ICC (1:500) |
| Antibody | AlexaFluor 568 goat anti-mouse IgG (H + L) (polyclonal, secondary antibody) | Invitrogen | REF A11004 | IHC/ICC (1:500) |

*Continued on next page*

Continued

| Reagent type (species) or resource | Designation | Source or reference | Identifiers | Additional information |
|---|---|---|---|---|
| Antibody | AlexaFluor 488 goat anti-guinea pig IgG (H + L) (polyclonal, secondary antibody) | Invitrogen | REF A11073 | IHC/ICC (1:500) |
| Chemical compound, drug | 6-Cyano-7-nitroquinoxaline-2,3-dione disodium salt hydrate (CNQX) | Sigma-Aldrich | Catalog # C239 | (10 µM) |
| Chemical compound, drug | D(−)-2-Amino-5-phosphonopentanoic acid (AP-5) | Sigma-Aldrich | Catalog # A8054 | (50 µM) |
| Chemical compound, drug | Tetrodotoxin (TTX) | Enzo Life Sciences | Catalog # BML-NA120-0001 | (1 µM) |
| Chemical compound, drug | DNase I | Sigma-Aldrich | Catalog # D5025 | (0.5 mg/ml) |
| Chemical compound, drug | Transferrin | Calbiochem | Catalog # 616,420 | (50 mg/500 ml) |
| Chemical compound, drug | Cytosine Arabinoside (Ara-C) | Sigma | Catalog # C6645 | |
| Chemical compound, drug | B-27 supplement | GIBCO | Catalog # 17504-010 | |
| Commercial assay, kit | ProFection Mammalian Transfection System | Promega | E1200 | |
| Chemical compound, drug | Matrigel | Corning | Catalog # 354,230 | (1:50) |
| Cell line (human kidney) | Human embryonic kidney-293 (HEK293) cells | ATCC | Catalog # CRL-1573; RRID: CVCL_0045 | |
| Strain, strain background | Sprague-Dawley rats, CD1 (Sprague-Dawley postnatal pups P2-3, M and F) | Charles River | Strain code: 400 | |
| Recombinant DNA reagent | Plasmid: pCI syn iGluSnFR | *Helassa et al., 2018* PNAS | Addgene_106,123 | Plasmid to transfect and express iGluSnFR |
| Software, Algorithm | Prism 8 | GraphPad | https://www.graphpad.com/ | |
| Software, Algorithm | Intellicount | *Fantuzzo et al., 2017* | N/A | |
| Software, Algorithm | Fiji | *Schindelin et al., 2012* | N/A | |
| Software, Algorithm | MATLAB. (2018). 9.7.0.1190202 (R2019b). | Natick, Massachusetts: The MathWorks Inc. | https://www.mathworks.com/products/matlab.html?s_tid=hp_products_matlab | |
| Software, Algorithm | easFRAP-web | *Koulouras et al., 2018* | https://easyfrap.vmnet.upatras.gr/?AspxAutoDetectCookieSupport=1 | |
| Software, Algorithm | MiniAnalysis | Synaptosoft | http://www.synaptosoft.com/MiniAnalysis | |
| Software, Algorithm | Clampfit | Molecular Devices | https://www.moleculardevices.com/ | |
| Software, Algorithm | Axopatch | Molecular Devices | https://www.moleculardevices.com/ | |
| Software, Algorithm | Vutara | Bruker: SRX Software | https://www.bruker.com/en/products-and-solutions/fluorescence-microscopy/super-resolution-microscopes/vutara-vxl.html | |

## Primary dissociated hippocampal neuron culture preparation

Primary hippocampal cultures were generated by dissecting hippocampi from P1–3 Sprague-Dawley rats as previously described (*Kavalali et al., 1999*), with some modifications. Briefly, dissected hippocampi were washed and treated with 10 mg/ml trypsin and 0.5 mg/ml DNAse at 37°C for 10 min.

Tissue was washed again, dissociated with a P1000 tip, and centrifuged at 1000 rpm for 10 min at 4°C. Cells were then resuspended and plated on Matrigel-coated 0 thickness glass coverslips in 24-well plates at a density of four coverslips per hippocampus. Cultures were kept in humidified incubators at 37°C and gassed with 95% air and 5% $CO_2$.

Plating media contained 10% fetal bovine serum (FBS), 20 mg/l insulin, 2 mM L-glutamine, 0.1 g/l transferrin, 5 g/l D-glucose, 0.2 /g $NaHCO_3$ in minimal essential medium (MEM). After 24 hr, plating media was exchanged for growth media containing 4 µM cytosine arabinoside (as well as 5% FBS, 0.5 mM L-glutamine, and B27) to inhibit glial proliferation. On days in vitro (DIV) 4, growth media was exchanged to a final concentration of 2 µM cytosine arabinoside. Cultures were then kept without disruption until DIV 14–21.

## Sparse neuron transfection

Neuronal transfections were performed on DIV 7 using a calcium phosphate kit (ProFection Mammalian Transfection System, Cat # E1200, Promega), based on a previously described method (*Sando et al., 2019*). Briefly, A precipitate was formed by mixing the following per each well in a 24-well plate: 1 µg of plasmid DNA, 2 µl of 2 M $CaCl_2$, and 13 µl $dH_2O$. This mixture was then added dropwise to 15 µl of 2× N-2-Hydroxyethylpiperazine-N'-2-Ethanesulfonic Acid (HEPES), while vortexing between drop addition. The precipitate was allowed to form for 15 min. Neuron conditioned media was saved and replaced with MEM and 30 µl of plasmid mixture was added dropwise to each well. Plates were returned to 5% $CO_2$ incubator at 37°C for 30 min. Then cells were washed twice with MEM, after which previously saved conditioned media was added back to each well. Neurons were analyzed at DIV 15–17 using a confocal microscope.

## Whole-cell patch clamp

Whole-cell patch clamp recordings were performed on pyramidal neurons at DIV 14–15 at a clamped voltage of −70 mV using a CV203BU headstage, Axopatch 200B amplifier, Digidata 1320 digitizer, and Clampex 9.0 software (Molecular Devices). Only experiments with <15 MOhm access resistance and <300 mA leak current were selected for recording.

Extracellular Tyrode solution contained (in mM): 150 of NaCl, 4 of KCl, 10 of D-glucose, 10 of HEPES, 2 of $MgCl_2$, 2 of $CaCl_2$ at pH 7.4 and 310–320 mOsm. The ~3–6 MΩ borosilicate glass patch pipettes were filled with the internal pipette solution contained the following (in mM): 115 Cs-MeSO$_3$, 10 CsCl, 5 NaCl, 10 HEPES, 0.6 ethylene glycol-bis(β-aminoethyl ether)-N,N,N',N'-tetraacetic acid (EGTA), 20 tetraethylammonium-Cl, 4 Mg-ATP, 0.3 Na$_3$GTP, and 10 QX-314 [*N*-(2,6-dimethylphenylcarbamoylmethyl)-triethylammonium bromide] at pH 7.35–7.40 and 300 mOsm.

To isolate excitatory currents, 50 µM APV and 50 µM picrotoxin (PTX, ionotropic GABA receptor inhibitor) were added to the bath solution. To isolate mEPSCs, 1 µM TTX sodium channel inhibitor, 50 µM PTX, and 50 µM D-AP5 were added. For EPSC recordings, the field stimulation was provided using a parallel bipolar electrode (FHC) immersed in the external bath solution, delivering 35 mA pulses (0.1 ms duration) via a stimulus isolation unit. Miniature events were identified with a 5-pA detection threshold and analyzed with MiniAnalysis (Synaptosoft, Fort Lee, NJ, USA).

## Live fluorescence imaging

Imaging experiments were done in Tyrode's buffer (as described above). Tyrode's solution containing either 0, 2, or 8 mM $Ca^{2+}$ with osmolarity titrated to 310–320 mosM, and 50 µM APV and 10 µM CNQX to prevent recurrent neuronal activity. Fluorescence was recorded using a Nikon Eclipse TE2000-U inverted microscope equipped with a ×60 Plan Fluor objective (Nikon, Minato, Tokyo, Japan), a Lambda-DG4 illumination system (Sutter Instruments, Novato, CA, USA) with FITC excitation and emission filters, and an Andor iXon + back illuminated EMCCD camera (Model no. DU-897E-CSO-#BV; Andor Technology, Belfast, UK). Images were acquired at 50 Hz to resolve fast spiking glutamatergic peaks. To induce photobleaching, the neutral density filter within the LAMDA-DG4 illumination system was removed in order to use 100% light intensity. This filter was reintroduced for subsequent live imaging of neurotransmisison after photobleaching.

Spontaneous activity was recorded over the course of 6–10 min. Evoked responses were elicited using a parallel bipolar electrode, delivering 35 mA pulses (0.1ms duration) at 5-s intervals. At the end

of each experiment, presynaptic boutons were visualized by delivering a high-frequency electrical stimulation (25 Hz 20 action potentials) or by perfusing 90 mM KCl in Tyrode's solution.

## Fluorescence analysis

Images were analyzed using Fiji (*Schindelin et al., 2012*). Local fluorescence maxima during 90 mM KCl stimulation were located using a custom macro and used to draw circular ROIs (of 2 µm diameter) around synapses. Fluorescence intensity over time was measured for each ROI and exported to Excel, along with the image metada containing treatment and stimulation time information. Data were analyzed using an unbiased method based on our previous studies (*Chanaday and Kavalali, 2018*). Briefly, background was subtracted linearly, and traces were smoothed at every three or five points. Spontaneous events were detected using a threshold of 3 standard deviations (SDs) above a moving average (baseline) of 4 s. Evoked events detection was time locked within 0.3 s of an AP delivery, at a threshold of 3 SD above baseline. Parameters including frequency, release probability, and amplitude were automatically estimated. All custom Matlab (Mathworks, Natick, MA, USA) scripts are available on Github (https://github.com/camilleswang/iGluSnFR-Analysis, copy archived at swh:1:rev:8f529b-60ce0561ad771dd553cc4f477dbbf7cfaf; *Wang, 2021*) and upon request.

## Fluorescence recovery after photobleaching

FRAP experiments were performed on an LSM 510 META confocal microscope (Carl Zeiss, Oberkochen, Germany) with a ×63 (NA1.4) objective. A small region ~1 µm in diameter was selected to be bleached. An unbleached reference region was selected to correct for artifactual photobleaching, and a background region was selected for normalizing the signal. Photoleaching was performed over four to five scans at 100% laser power. Fluorescence recovery data were run through easyFRAP-web (https://easyfrap.vmnet.upatras.gr/), a web-based tool for the analysis of FRAP data, which provides photobleaching depth, gap ratio, normalization data, and curve fitting parameters (*Koulouras et al., 2018*). Values were computed with a double normalization method (using a neighboring unbleached area as control) and a single exponential equation curve fitting. Immobile fractions were calculated from the asymptote of the single exponential equation (for detailed info please see *Koulouras et al., 2018*).

## Immunofluorescence

At DIV 16–17, neuron cultures were fixed with 4% paraformaldehyde (PFA) and 4% sucrose in phosphate-buffered saline (PBS) at room temperature for 15 min. After three washes, cells were permeabilized for 30 min with 0.2% Triton-X in PBS. Following another three washes, blocking solution consisting of 1% bovine serum albumin (BSA) and 2% goat serum was added for 1–2 hr. Primary antibody diluted in blocking solution was added and incubated overnight at 4°C in a humid chamber. Primary antibody against MAP2 (1:500) was used to detect neuronal architecture, anti-vGluT1 (1:500) was used to detect presynaptic boutons, and anti-PSD95 (1:200) was used to detect postsynaptic specifications. The following day, coverslips were washed three times then incubated with species-appropriate Alexafluor secondary antibodies at 1:500 for 60–90 min at room temperature. Coverslips were then washed and mounted on glass slides, and they were imaged using an LSM 510 META confocal microscope (Carl Zeiss, Oberkochen, Germany) with a ×63 (NA1.4) objective.

## Super resolution microscopy

Neuronal cultures were grown on 1.5 glass bottom Mattek dishes (Cat # P35G-1.5-14-CGRD), and they were transfected with iGluSnFR at DIV7. At DIV 16–17, neuron cultures were fixed and permeabilized similar to immunofluorescence experiments. Cells were then washed and blocked with 1% BSA, 2% goat serum, and 2% donkey serum for 2 hr at room temperature. Primary antibody against GFP (rabbit, Synaptic Systems, 1:300) was added to detect iGluSnFR molecules overnight, as well as PSD95 (mouse, Synaptic Systems, 1:100). The following day, primary antibody was washed off and cells were incubated in secondary antibody (AlexaFluor 647 at 1:500; AlexaFluor 568 at 1:500) for 1.5 hr at room temperature. Secondary antibodies were then washed off and cells were postfixed with 4% PFA for 10 min to enable long-term storage of samples in PBS.

For STORM, the extracellular imaging buffer was made fresh on ice prior to imaging (50 mM MEA, 1× glucose oxidase, 1× catalase, 4% 2-mercaptoethanol in Buffer B, which is comprised of 50 mM

Tris–HCl + 10 mM NaCl + 10% glucose). TetraSpeck beads (100 nm; Invitrogen) mounted on a glass coverslip were used to calibrate alignment between the two channels. Imaging was performed on a Vutara VXL Microscope from Bruker. All data analysis was performed with the Vutara software, using the spatial distribution module and cluster analysis module to measure density values and detect clustering of iGluSnFR tagged by GFP. The Vutara software finds the geometrical center of the cluster to run a pair correlation analysis.

## Statistical analysis

Data in graphs were presented as mean ± standard error of the mean (SEM) unless indicated otherwise. Sample sizes were stated in the figure legends and represented as the number of coverslips, unless otherwise indicated. Statistics were done on the averages of coverslips, rather than individual synapses to avoid falsely significant results due to very large sample sizes (the number of synapses and release events can run in the thousands). Individual synaptic values are represented as denoted in several graphs to demonstrate the distribution of values. Sample sizes were based on previous studies in the field of molecular and cellular neuroscience as opposed to using statistical methods prior to experimentation. To ensure reproducibility, each set of experiments were performed across multiple coverslips in at least two sets of cultures. Vutara Software Analysis was used to perform the pair correlation analysis of the super resolution images using the spatial distribution module and cluster analysis module, as well as to detect clustering of iGluSnFR tagged by GFP. GraphPad Prism was used to perform the statistical analyses of all other sets of experiments.

A Welch's $t$-test was used to compare effects in pairwise datasets obtained from synapses or neurons under distinct conditions. A Kolmogorov–Smirnov test was used to compare the cumulative histogram of two groups. A chi-squared test was used to analyze the correlation between two groups. For parametric analysis of multiple comparisons, two-way analysis of variance (ANOVA and one-way ANOVA) with Tukey post hoc analysis were used. Outliers were identified with Robust regression and Outlier removal (ROUT) method. Differences among experimental groups were considered statistically significant when a p value ≤0.05 was reached. Specifics of statistical tests and p value denotations are listed in figure legends.

## Acknowledgements

We would like to thank current and former members of Kavalali and Monteggia laboratories for numerous invaluable discussions. This work was supported by the National Institute of Health grants MH66198 (ETK), GM007347 (CSW), MH081060 and MH070727 (LMM), and NARSAD Young Investigator Award (NLC).

## Additional information

### Competing interests

Lisa M Monteggia: Reviewing editor, *eLife*. The other authors declare that no competing interests exist.

### Funding

| Funder | Grant reference number | Author |
|---|---|---|
| National Institute of Mental Health | MH66198 | Ege T Kavalali |
| National Institute of Mental Health | MH081060 | Lisa M Monteggia |
| National Institute of Mental Health | MH070727 | Lisa M Monteggia |
| National Institute of General Medical Sciences | GM007347 | Camille S Wang |

| Funder | Grant reference number | Author |
| --- | --- | --- |
| Brain and Behavior Research Foundation | | Natali L Chanaday |

The funders had no role in study design, data collection, and interpretation, or the decision to submit the work for publication.

## Author contributions

Camille S Wang, Conceptualization, Data curation, Formal analysis, Investigation, Methodology, Software, Validation, Writing - original draft, Writing - review and editing; Natali L Chanaday, Data curation, Formal analysis, Funding acquisition, Investigation, Methodology, Validation, Writing - original draft, Writing - review and editing; Lisa M Monteggia, Funding acquisition, Project administration, Supervision, Writing - review and editing; Ege T Kavalali, Conceptualization, Funding acquisition, Project administration, Supervision, Validation, Writing - review and editing

## Author ORCIDs

Camille S Wang ![ORCID] http://orcid.org/0000-0002-2178-5754
Natali L Chanaday ![ORCID] http://orcid.org/0000-0002-3376-5187
Lisa M Monteggia ![ORCID] http://orcid.org/0000-0003-0018-501X
Ege T Kavalali ![ORCID] http://orcid.org/0000-0003-1777-227X

## Ethics

Animal procedures conformed to the Guide for the Care and Use of Laboratory Animals and were approved by the Institutional Animal Care and Use Committee at Vanderbilt University School of Medicine (Animal Protocol Number M1800103).

## Decision letter and Author response

Decision letter https://doi.org/10.7554/eLife.76008.sa1
Author response https://doi.org/10.7554/eLife.76008.sa2

## Additional files

### Supplementary files

• Transparent reporting form

### Data availability

All data generated or analysed during this study are included in the manuscript and supporting files; Source Data files have been provided for all Figures and Figure Supplements. The custom Matlab script used to analyse the data is deposited in GitHub (https://github.com/camilleswang/iGluSnFR-Analysis, copy archived at swh:1:rev:8f529b60ce0561ad771dd553cc4f477dbbf7cfaf) and is freely available.

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
