## [Editor Report]

Chemical neurotransmission is the major form of inter-neuronal communication in the CNS. The classical model is that the pathways utilized by action potential-evoked and spontaneous neurotransmission are the same. Multiple lines of evidence now suggest that the Venn diagrams describing the vesicle identity, regulatory systems, and postsynaptic receptors utilized by evoked and spontaneous transmission, overlap incompletely. In this paper, Wang and colleagues, explore the distribution of glutamate release sites at hippocampal synapses using a genetically encoded glutamate sensor and photobleaching, and their results indicate that evoked release is confined to a smaller region of each excitatory synapse than the more dispersed spontaneous release.

---

## [Decision Letter]

**Decision letter after peer review:**

Thank you for submitting your article "Probing the segregation of evoked and spontaneous neurotransmission via photobleaching and recovery of a fluorescent glutamate sensor" for consideration by *eLife*. Your article has been reviewed by 3 peer reviewers, one of whom is a member of our Board of Reviewing Editors, and the evaluation has been overseen by John Huguenard as the Senior Editor. The following individuals involved in review of your submission have agreed to reveal their identity: Chris G Dulla (Reviewer #2); Michael Hoppa (Reviewer #3).

Essential revisions:

Overall, the three reviews were consistent and positive about the submission. As you will see below, they felt the work addressed an important question using an interesting approach. However, all reviewers felt that substantial modifications are required to clarify the conclusions and make the work accessible to a general readership of *eLife*. Rather than cut and paste from the lists provided by the referees I have identified three major types of problem in the list below. However, I point out that it is essential that all of the specific points mentioned in the attached reviews should be addressed.

1) Frequently the reader has to make assumptions about what is being tested and why. The introduction to each set of experiments needs to be more explicit and include sufficient explanation of the hypothesis being tested and the rationale for the approach. In addition, there is often a deficiency of key experimental details such as the frequency of stimulation or nature of a region of interest forcing the reader to guess what was done.

2) The reviewers have suggested only a few additional experiments and suspect that these may have already been performed. Examples are included in questions 12 and 20 (reviewer 1).

3) The results clearly benefit from the high resolution imaging that can be obtained in dispersed cultures, and the preparation should be detailed in the title or abstract. Is the authors' approach feasible in a more intact system, such as in vivo or acute brain slices? In lieu of such data, the discussion should address how synaptic structure/function may be different in culture than in the intact brain and how this might affect the conclusions.

*Reviewer #1 (Recommendations for the authors):*

This study addresses an interesting and important question using a novel approach. In order to increase the impact of the work I have a number of suggestions about additional experiments that may clarify issues brought up by the manuscript. I suspect that the authors may have already performed some of these. In addition, I believe that the authors should substantially revise sections of the results to make the work accessible to the potential readers. I found it quite difficult to understand the rationale behind the order and design of experiments at times. I suspect that the authors will be able to improve the manuscript substantially by being more explicit about some of the experimental conditions and the hypotheses being tested. For instance, explicit mention of the alternate hypothesis that the same glutamate receptors are involved in spontaneous and evoked release may help the reader. Major required changes are listed below:

1. In Figure 1 the authors include live cell imaging and SRM on fixed tissue. Please indicate explicitly in the text the different conditions under study. Direct imaging of the iGluSnFR in live versus secondary antibody to GFP in fixed?

2. Is sparse transfection important? Please define and state why.

3. L 95. There are no circles on Figure 1B. Is this the best time to mention them?

4. L 95 Please define high-frequency stimulation at this point so that the reader can understand the experiment.

5. I believe figure 1B shows the fluorescence signal accumulated during or after a period of stimulation. Please clarify and consider that it may be helpful to see a control image before stimulation and after exposure to bath application of glutamate to clarify nonuniform distribution of probes.

6. L100 Please clarify if the pair correlation analysis was how you determined that the probe expression was mainly synaptic. Citation to support the use of pair correlation analysis.

7. I found the second paragraph in results very difficult to follow and wonder if this could be rewritten to improve clarity. I was unclear how the normalization to the dendritic marker was performed. Citations to support this approach.

8. Figure 1C calibration bar required. Please provide the corresponding DIC and PSD 95 images as well.

9. Supplementary figure 1B, Y axis label should be revised for dendritic marker signal and units should be per square microns.

10. Line 129 please describe more clearly how release probability was calculated.

11. L 143. Please discuss if there are other possible reasons that the size of the events was increased substantially by high extracellular calcium and why the citations suggest multi-vesicular release and spillover is likely.

12. Figure 2 D is this massive increase in event size observed in 8 mM an outlier or consistent with average? It appears there is a 5 to 10 fold increase in the size of each event. Have the authors attempted to evaluate if such a large amount of spillover is feasible. For instance, do experiments examining a small region of interest using pressure application of glutamate suggest a dynamic range of detection consistent with the suggestion of spillover.

13. L165 please clarify why comparable amplitudes for evoked spontaneous events indicate single vesicle release.

14. Line 170 please explain why seeing QNX and APV was used in these experiments.

15. L181. Explain why specifically 100 mM sucrose was utilized in this experiment.

16. Figure 3. Do the averaged spontaneous events have similar kinetics to mEPSCs?

17. L223-227. Any citations to support these statements?

18. L234 Figure 4G there seem to be some possible differences between HEK cells and neurons. The fractional bleaching, fractional recovery, and rate of recovery seem much larger in the HEK cell exemplar. How were the measurements included in 4H and I made? Was it by fitting an exponential and using the asymptote to estimate the immobile fraction?

19. Figure 4H, Y-axis label is missing.

20. L269-279. I am surprised and puzzled by the failure of electrical stimulation to influence the rate of photo bleaching. In addition to spillover the authors may wish to explicitly mention why this does not support the idea that receptors involved in spontaneous and evoked release are the same. It seems unlikely that spillover from spontaneous release during this brief time is adequate to activate all of the receptors involved in evoked release. Additional experiments to clarify this finding seem key. Would use of high bath concentrations of calcium increase evoked release sufficiently so that brief periods of stimulation could be tested. Presumably evoked release would be increased proportionately more than spontaneous release, so that less bleaching may occur in the control experiment sounds stimulation.

*Reviewer #2 (Recommendations for the authors):*

Overall, this is an exciting and potentially very impactful study. The work presented is of very high quality and there is great interest in understanding spontaneous vs evoked release. Because this study is extremely technical in nature, there are a number of technical and methodological points that need some additional clarification and consideration to properly interpret the findings.

– An interesting and central finding of this study is that iGluSnFR appears to be concentrated at synapses. This seems counter intuitive. Synaptic real estate is so in demand and many molecules have specific targeting mechanisms to bring them to the synapse. How does iGluSnFR end up concentrated at the synapses without a specific synapse targeting mechanism? Is this finding solely based on imaging data gathered during high frequency stimulation? If so, one could argue that GluSnFR is not enriched in synapses but is just highly active during intense neuronal activity. Some clarification on GluSnFR protein vs GluSnFR signal localization would improve this argument.

– Second similar question – does the synaptic enrichment of GluSnFR fully recover after photobleaching and if so on what timescale? Because GluSnFR uses a DF/F approach to detect glutamate, it would be helpful in interpreting the evoked vs spontaneous FRAP recovery data to see the recovery of basal fluorescence (as in Figure 1C, expressed in AU) vs the ability to detect glutamate (as in the rest of the figures, expressed in DF/F).

– How was the mobile vs immobile fraction of iGluSnFR calculated? It's not clear from the text exactly how this ratio was arrived at.

– The photobleaching experiments used a 1 μm diameter photobleaching area. ROIs used for quantifying GluSnRF activity in the synapse used a 2-3 μm diameter ROI. If the photobleaching was centered over the active zone of the synapse, it stands to reason that the region associated with evoked release (central within the synapse) and the region associated with spontaneous release (peripheral within the synapse) are not equally photobleached. That could explain some of the data and should be considered as a potential caveat. Have the authors considered using a larger area of photobleaching to completely bleach everything in the synaptic ROI?

– In Figure 1 C it would be helpful to show PSD95. The figure legend says that the particles shown colocalize with PSD95 but there is no way to interpret or evaluate that with the current figure presentation.

– Because the central focus of the paper is spontaneous synaptic release, it would help the manuscript to give the reader a better description of exactly how spontaneous release events were detected. In the paper the authors state "we examined parameters of spontaneous iGluSnFR events" but it's not clear how they got those parameters to evaluate. Were the exact same ROIs used? Was there a signal to noise threshold used to identify events from noise? Were spontaneous events captured before or after evoked? Was that switched to control for the potential effects of stimulation of spontaneous release?

– It is not really clear why the authors did not use TTX throughout their experiments to analyze spontaneous release. Using TTX is the standard in the field for isolating action-potential independent events. I see that it allows them to do their stimulation experiments in the same cultures as their evoked experiments, which is essential to their study, but it leaves open the possibility that some spontaneous events are AP-associated. Have the authors made recordings from neurons in glutamate synaptic blockers and shown that 100% of the AP are absent? That would be a helpful control to validate their method.

– GluSnFR detection of evoked release is suppressed for hours after photobleaching while the ability to detect spontaneous release using GluSnFR recovers much more rapidly. This is a very clear and exciting finding! An implication of this result, however, is that 100% of the events that contribute to evoked release are spatially compartmentalized from any of the machinery that drives the ability to detect spontaneous release. Spontaneous release is detected using the same ROIs as evoked release (as far as I can tell), so that means within an ROI spontaneous release is being detected while evoked is not. Data presented shows that glutamate release is happening in both situations, but just can't be detected in the evoked release situation due to the photobleaching. So evoked release is somehow spatially restricted to the GluSnFR molecules in the synapse and cannot be detected by other GluSnFR molecules that detect spontaneous release in the same ROI. This is hard to reconcile with the fact that glutamate diffusion is likely the driving factor by which glutamate is cleared from the center of synapses.

*Reviewer #3 (Recommendations for the authors):*

This paper by Wang CS et al., investigates the localization of spontaneous and evoked vesicle fusion within a synapse. This is an important question and an interesting central finding. The authors nicely document the use of a glutamate-sensitive fluorescent reporter (iGluSnFR) to measure spontaneous and evoked vesicle fusion. They do a great job of characterizing iGluSnFR and take advantage of the probe's rapid bleaching to follow up previous molecular and physiological characterization of spontaneous vesicle fusion with novel findings of unique spatial segregation of spontaneous vesicle fusion. This finding is mainly based on this bleaching characteristic of GluSnFR. I was impressed by the idea of using the bleaching of a membrane probe to resolve the location of vesicle fusion. I have some questions and comments that I believe are important to be addressed to better understand the findings and their interpretation for an accepted manuscript.

1. iGluSnFR has a very high affinity to provide it with the sensitivity for detecting single vesicle fusion. A concern would be if the sensor is also detecting vesicle fusion as spillover from adjacent boutons of untransfected neurons undergoing spontaneous release. This concern is somewhat mitigated since the rate of spontaneous release is not much higher than expected from electrophysiology and good criteria are deployed for detection of events vs noise. However given that location is so important for the interpretation, these "spill-over" events could appear to be released from unique discrete areas of the synaptic bouton but are in fact coming from nearby synapses and the diffusing glutamate is being detected. A better characterization of the location of where spontaneous events were detected optically in the synapse relative to evoked might help resolve this concern. Were the ROIs formed from high frequency stimulation (or high potassium?) also in the same location for measuring spontaneous release? Were spontaneous vesicle fusions also detected outside of synapses identified by stimulation that might suggest spillover or solely restricted to regions with evoked release identified on dendrites?

2. The authors lead me to believe that GluSnFR bleaching is primarily dependent on activation of the indicator by glutamate binding. The difference in bleaching from glutamate perfusion compared to during rest or "no stimulation" for selectively impairing detection of evoked vs spontaneous release is striking and really interesting. That being said, it is very hard to understand the explanation the authors provide for the lack of difference between unstimulated and stimulated bleaching conditions. A difference here seems like the most direct experimental test to define separate release sites for evoked and spontaneous. If I understand the authors correctly, the lack of difference is the result of the spontaneous release of glutamate that contaminates this result. However, spontaneous release has a very low rate and would be releasing very small amounts of glutamate over the bleaching times identified. During 2 minutes of photobleaching when a very acute difference is found the rate of 0.1 Hz for spontaneous release the authors measured would mean that synapses experience on average 1 spontaneous vesicle fusion event during 2 minutes of photobleaching. This one release event would not be localized to sites of evoked release as the authors interpret the data yet would cause substantial bleaching equivalent to several vesicles fusing during stimulation. As it stands, 30 APs delivered during standard imaging over 2.5 minutes don't seem to cause any bleaching or decrease in detection efficiency (Figure 5A). To better understand the amount of bleaching that occurs in the three conditions, it would be useful to know both the difference in the illumination intensity between standard measurements and during photobleaching as well as the changes in resting fluorescence iat the synapse/ROIs. Presumably at the bleaching laser illumination power the same delivery of 30 APs @ 0.2 Hz (2.5 minutes) would rapidly decrease to very low levels if this interpretation of rapid bleaching when glutamate in present is correct. Otherwise, bleaching of GluSnFR doesn't require glutamate and perhaps another unknown property of local GluSnFR diffusion within a spine that the authors mention is responsible for the lack of difference in bleaching sensitivity to boutons at rest vs stimulation. This could still be in line with the authors' interpretation of unique release sites for evoked neurotransmission, but less easy to understand than selective loss from use of evoked release sites by glutamate dependent photobleaching. Perhaps lowering the "bleaching" laser intensity during stimulation would ablate/bleach detection of evoked release during stimulation that also preserves spontaneous release detection.

[Editors’ note: further revisions were suggested prior to acceptance, as described below.]

Thank you for resubmitting your work entitled "Probing the segregation of evoked and spontaneous neurotransmission via photobleaching and recovery of a fluorescent glutamate sensor" for further consideration by *eLife*. Your revised article has been evaluated by John Huguenard (Senior Editor), a Reviewing Editor and a reviewer.

The manuscript has been improved but there are some important issues that need to be addressed, (see below). The major changes required, focus on the re-organization and expansion of the manuscript to clarify the results reported by the authors in this innovative study. In addition to the recommendations below, it is important that the authors substantially expand the discussion about whether there might be differences in iGluR activation in the intact nervous system. They address this very briefly in the current discussion (lines 546- 552). Please expand this section and provide some citations to justify the rationale for the validity of the approach.

*Reviewer #1 (Recommendations for the authors):*

This resubmission describes work using a genetically encoded glutamate sniffer to study synaptic transmission in hippocampal cultured neurons. It investigates the properties of evoked and spontaneous transmission and identifies differences in the rates of photobleaching suggesting that the receptors involved in these 2 signaling pathways are localized differently. The authors have made a number of changes to the initial submission that have improved the manuscript, however, there are a number of remaining problems that need to be addressed before it is suitable for publication in Life.

Although much improved, the manuscript may be easier to read if the authors began the Results section by describing the distribution of iGluSnFR and how this compares with that for vGlut1 and PSD 95 distribution to indicate that the glutamate sniffer is found near to synapses (Figure 1 Suppl 2). They could then move on to describing iGluSnFR localization via super-resolution microscopy and then show how excitatory puncta number are similarly impacted by transfection with GFP, the sniffer, and MAP2.

The authors should also improve the description of their approach to analyzing the experiments that employed super-resolution microscopy. They describe experiments in which PSD 95 and GFP staining is identified in fixed issue and how they use the DB scan method to identify that this staining is clustered for both PSD and GFP. Some of these sites colocalize. They then go on to use pair correlation to examine the clustering of the iGluSNFR signal and show that it is not uniformly distributed within each of the clusters but highly concentrated close to its centre. It is unclear to me what criteria the authors use to justify describing this as the centre of the synapse (line 111).

In an earlier review, (question 5) I asked to see a control image from before stimulation of old Figure 1B. The summation projection is now in figure 2A but there is still no control image to illustrate how the fluorescence staining of iGluSNFR increases. Perhaps the authors have a control image for this, or another similar experiment, that could be included in this figure or a supplementary figure.

On line 155 of the manuscript the authors describe using regions of interest of 2 µm in diameter. However, in the methods section they mention ROIs were 2 to 3 µm in diameter. Please clarify if there is a discrepancy or the methods are referring to different experiments. Also, please mention if the fluorescence signal captured from the region of interest the sum, average or maximum values from the pixels therein?

Line 222, can the authors provide a citation confirming that the antagonists CNQX and APV do not interact with iGluSNFR?

Line 287, can the authors confirm that it is only in the experiments testing probe mobility that they used a 1 µm diameter region of interest for bleaching purposes. In all later experiments, using bleaching as a tool did they photobleach the entire field of view and were these same experiments performed using an EM CCD. Please clarify these details in the manuscript. Similarly, is it the case that in the late experiments shown in figure 5 and associated supplementary figure, that the regions of interest were always identified by activity during the evoked phase of the experiment, or were some regions of interest identified during the spontaneous phase? This information could be confirmed in the text to emphasize to the reader that the two segregated detection systems for spontaneous and evoked release are tightly associated.

Rather surprisingly the authors find the rate of bleaching of evoked signals is unchanged by stimulation during the bleaching. Given that there is no iGluSNFR response, I do not understand the hypothesis advanced concerning low probability of release and unbleached responsive sites mentioned on lines 356-358. The data appear to show that bleaching was saturated after a few minutes of photobleaching with no later responses to stimulation as mentioned in the hypothesis.

However, I think the experiments using high calcium follow-up are interesting. It seems that the high calcium is reducing the photobleaching in the lack of stimulation compared to the low calcium experiments (Figure 5 supp 1 G). I wonder if the results could also be explained by reduced action potential firing due to reduced excitability in the presence of high calcium. In addition, the bleaching is accelerated in the spontaneous release experiments using high calcium. Presumably this is because of increased rates of spontaneous release. These possibilities should be included in the discussion.

These differences reported in the rate of recovery of glutamate signaling, vividly underline the authors' thesis that movement of iGluSF was differentially impacted at sites that detect spontaneous and evoked release. Their major result showing that detection of evoked and spontaneous release detected in a 2 µm region of interest are so differently affected following bleaching, supports the authors' ideas that there is separation between the postsynaptic pathways mediating evoked and spontaneous release.

Have the authors estimated what fraction of excitatory synapses in their cultures express the iGluSnFR and how much of the iGluSnFR are not clustered at synapses? The first part of this question may be of interest to others and the second part may help identify the source for the iGluSnFR that reverses the bleaching as the neurons recover. If there was a wide field of bleaching and most of the iGluSnFR is clustered at synapses then one imagines that there is little expressed iGluSnFR to diffuse to the affected areas and maybe new protein formation may be required. Is it possible that the iGluSnFR at evoked sites is more sensitive to bleaching compared to that involved in mini detection and that contributes to the differences observed?

*Reviewer #3 (Recommendations for the authors):*

As before, the findings of this manuscript are very interesting. The authors addressed many of my concerns with the initial manuscript and have now included the necessary experimental details. I believe that the measurements and experiments conducted in this paper can now be made by other groups and the results understood to support the conclusions of the authors. I believe that supplementary experiments throughout are useful.

Figure 5 still seems to me to be a critical experiment for the interpretation and I found two parts a little unclear in the revisions, and could benefit from additional edits.

Line 354-355: "One explanation is basal activity…." What do you exactly mean by basal activity? Does this mean spontaneous release or action potentials and evoked release occurring independently of your stimulation protocol? From the methods I was under the impression that CNQX and APV were present to block electrical activity outside of field stimulation, so I wanted to be sure you were suggesting spontaneous release, but expansion here would be helpful to know exactly what the authors mean.

Line 359-360

"To test the latter hypothesis, we repeated the experiments in a higher extracellularca^2+^ of 8 mM. In this condition, probability of release is close to 1, and most synaptic sites should be activated and photobleached."

It would be more helpful to include the exact changes in Pr from the paper in these calcium conditions, shifting from (~ 0.5 and 0.9). I could not find the exact number in Figure 2 or the manuscript but estimated from the bar graphs.

Otherwise, the changes were very helpful and made the manuscript more compelling.

---

## [Author Response]

1) Frequently the reader has to make assumptions about what is being tested and why. The introduction to each set of experiments needs to be more explicit and include sufficient explanation of the hypothesis being tested and the rationale for the approach. In addition, there is often a deficiency of key experimental details such as the frequency of stimulation or nature of a region of interest forcing the reader to guess what was done.

We thank the reviewers for the helpful and constructive comments. We have included further details throughout the manuscript to better state the scientific question, the hypothesis being tested, and expanded the explanation of rational experiments to better flow with the experiments performed, the results and conclusions.

2) The reviewers have suggested only a few additional experiments and suspect that these may have already been performed. Examples are included in questions 12 and 20 (reviewer 1).

We have addressed these few additional experiments in the revised paper (see Supp. Figures 1, 3I-K).

3) The results clearly benefit from the high resolution imaging that can be obtained in dispersed cultures, and the preparation should be detailed in the title or abstract. Is the authors' approach feasible in a more intact system, such as in vivo or acute brain slices? In lieu of such data, the discussion should address how synaptic structure/function may be different in culture than in the intact brain and how this might affect the conclusions.

We now mention the preparation of cultured neurons in the abstract. In the Discussion section, we now comment on the feasibility of our approach in tissue systems and how the findings might be translatable to the intact brain as follows.

“While we would expect the findings from primary cultured neurons to largely translate to an intact brain system, there may be some differences. For instance, the differences in extracellular matrix composition may affect diffusional properties, and the 3D nature of synapse organization within the dense neuropil could lead to differences in glutamate dynamics compared to a monolayer of neurons. It is also possible that more complex ex vivo brain tissues may introduce technical complications in detecting differences in photobleaching and fluorescence recovery of iGluSnFR probes.”

Reviewer #1 (Recommendations for the authors):This study addresses an interesting and important question using a novel approach. In order to increase the impact of the work I have a number of suggestions about additional experiments that may clarify issues brought up by the manuscript. I suspect that the authors may have already performed some of these. In addition, I believe that the authors should substantially revise sections of the results to make the work accessible to the potential readers. I found it quite difficult to understand the rationale behind the order and design of experiments at times. I suspect that the authors will be able to improve the manuscript substantially by being more explicit about some of the experimental conditions and the hypotheses being tested. For instance, explicit mention of the alternate hypothesis that the same glutamate receptors are involved in spontaneous and evoked release may help the reader. Major required changes are listed below:

We thank the reviewers for the helpful and constructive comments. As stated above, we have included further details throughout the manuscript to better state the scientific question, the hypothesis being tested, and expanded the explanation of rational experiments to better flow with the experiments performed, the results and conclusions.

1. In Figure 1 the authors include live cell imaging and SRM on fixed tissue. Please indicate explicitly in the text the different conditions under study. Direct imaging of the iGluSnFR in live versus secondary antibody to GFP in fixed?

We now explicitly indicate that iGluSnFR detection via STORM occurs in fixed hippocampal cultures stained with anti-GFP antibodies, followed by fluorescent secondary antibodies, since GFP is not a suitable fluorophore for STORM (i.e. GFP does not exhibit fast photoswitching cycles). We have also moved the live imaging methodology (previously in Figure1B) to a later section of the Results, to avoid confounding the fixed and live imaging experiments (live imaging method is now in Figure 2A).

2. Is sparse transfection important? Please define and state why.

Since iGluSnFR is expressed at the plasma membrane, it renders the whole neuron surface fluorescent. Thus, sparse transfection of only a small number of neurons in the culture allows for clear identification and measurement of single synapses with very low background fluorescence, which would be almost impossible if all neurons were fluorescent. We have added a clarifying statement in the corresponding Results section to explain this point.

3. L 95. There are no circles on Figure 1B. Is this the best time to mention them?

We have changed the arrows noting synapses in Figure 1B (now Figure 2A) into circular regions of interest to improve the clarity of the analysis.

4. L 95 Please define high-frequency stimulation at this point so that the reader can understand the experiment.

We have added a brief phrase explaining that high frequency stimulation is the delivery of 25 APs at 20 Hz.

5. I believe figure 1B shows the fluorescence signal accumulated during or after a period of stimulation. Please clarify and consider that it may be helpful to see a control image before stimulation and after exposure to bath application of glutamate to clarify nonuniform distribution of probes.

Figure 1B, and all of the STORM imaging, was done by immunolabeling the GFP molecule on iGluSnFR and staining with secondary antibodies specific for STORM. Thus, we are not using the fluorescence of iGluSnFR itself but rather labeling all of the iGluSnFR proteins, and not just the ones activated by glutamate. We have clarified this in the revised version of the manuscript.

6. L100 Please clarify if the pair correlation analysis was how you determined that the probe expression was mainly synaptic. Citation to support the use of pair correlation analysis.

We have clarified the methodology (DBSCAN) we used to determine that the probe expression was largely synaptic and added a corresponding reference.

7. I found the second paragraph in results very difficult to follow and wonder if this could be rewritten to improve clarity. I was unclear how the normalization to the dendritic marker was performed. Citations to support this approach.

We have revised this section to improve the clarity, as well as added citations to previous work supporting our approach and interpretation.

8. Figure 1C calibration bar required. Please provide the corresponding DIC and PSD 95 images as well.

We have added the calibration bar into the figure and also included the PSD95 images in Supplementary figure 1A-B. Unfortunately, we cannot obtain DIC images with our STORM imaging system and thus these are not included.

9. Supplementary figure 1B, Y axis label should be revised for dendritic marker signal and units should be per square microns.

We have made this correction.

10. Line 129 please describe more clearly how release probability was calculated.

We used failure analysis to estimate release probability, which is now explained in the revised manuscript.

11. L 143. Please discuss if there are other possible reasons that the size of the events was increased substantially by high extracellular calcium and why the citations suggest multi-vesicular release and spillover is likely.

We have expanded the discussion of why fusion of multiple vesicles and spill over are the most likely reasons leading to increased amplitude of iGluSnFR events at high extracellular calcium, including supporting references from our colleagues.

“Based on the literature, this could be due to release of multiple vesicles as release probabilities increase at higher ca^2+^ concentrations (Leitz and Kavalali, 2011; Rudolph et al., 2015), or glutamate spill over from adjacent sites (Armbruster et al., 2020), or a combination of these factors.”

12. Figure 2 D is this massive increase in event size observed in 8 mM an outlier or consistent with average? It appears there is a 5 to 10 fold increase in the size of each event. Have the authors attempted to evaluate if such a large amount of spillover is feasible. For instance, do experiments examining a small region of interest using pressure application of glutamate suggest a dynamic range of detection consistent with the suggestion of spillover.

We thank the reviewer for this insightful comment. The reviewer is correct, we inadvertently had chosen an outlier for our representative trace. We apologize for this oversight and have removed it and replaced it with one that shows the average evoked changes closer to the ~2x increase in amplitudes (now Figure 2E), as demonstrated in Figure 2I. We have also further elaborated on our rationale for the increased event amplitude at higher calcium concentrations. The release of multiple vesicles has been found across many synapse types in the brain, including excitatory hippocampal synapses, and this form of release is increased when release probabilities are increased (i.e. with increased extracellular calcium). Glutamate diffusion, as modeled by Armbruster et al., (2020), can occur up to 2 μm from the center of the synapse. While puffing glutamate may be useful, it would be difficult to recapitulate the precise manner by which glutamate is released, such as within nanocolumns at evoked release sites, which is critical for the differential effects we observe with photobleaching.

13. L165 please clarify why comparable amplitudes for evoked spontaneous events indicate single vesicle release.

We thank the reviewer for this insightful comment. At the single synapse level, evoked release is a binary process that is due to the low probability of release. Both evoked and spontaneous neurotransmission involve the release of a single synaptic vesicle at individual synapses. Each vesicle should contain similar amounts of neurotransmitters, which would activate similar numbers of glutamate fluorescent probes; thus, measured amplitudes should be comparable. We have added this explanation to the Results section.

14. Line 170 please explain why seeing QNX and APV was used in these experiments.

CNQX and APV block AMPARs and NMDARs, respectively, and were used to block recurrent network activity in the cultures. We have expanded on this explanation in the revised Results section.

15. L181. Explain why specifically 100 mM sucrose was utilized in this experiment.

While 300-500 mOsm sucrose leads to fusion of the complete RRP, 100 mOsm only causes stochastic fusion that allows the resolution of single release events, which is why we used it. The explanation was included in the corresponding section of the Results.

16. Figure 3. Do the averaged spontaneous events have similar kinetics to mEPSCs?

The average rise and decay times of mEPSCs are in the order of 0.5 and 5 ms, respectively. For iGluSnFR spontaneous events, the rise and decay times are in the order of 3 and 15-20 ms, respectively, an order of magnitude slower than mEPSC (values come from experiments done in our laboratory, also see Helassa et al., PNAS 2017). Spontaneous events measured using iGluSnFR do not have the same kinetics as mEPSCs.

17. L223-227. Any citations to support these statements?

We have added citations to support these statements. See below for changes:

“Immobile fractions reflect a subpopulation of probes that cannot be replenished by non-bleached probes. The existence of this immobile set of fluorophore molecules could be due to the geometry of the neuronal structure, for instance due to limitation of diffusion by surface proteins (Chen et al., 2021; Tardin et al., 2003). Another explanation could be that this is an intrinsic property of the probe, in which it interacts with proteins that tether it to the membrane surface.”

18. L234 Figure 4G there seem to be some possible differences between HEK cells and neurons. The fractional bleaching, fractional recovery, and rate of recovery seem much larger in the HEK cell exemplar. How were the measurements included in 4H and I made? Was it by fitting an exponential and using the asymptote to estimate the immobile fraction?

We agree with the reviewer that there is a tendency to a higher immobile fraction in HEK cells, but due to variability it is not statistically significant. To analyze the FRAP experiment we used a free web-based software that has been widely validated by other labs (see Koulouras et al., 2018). As the reviewer pointed out, the recovery time constant was obtained by a single exponential fitting and the immobile fraction by subtracting the asymptote from the baseline fluorescence.

19. Figure 4H, Y-axis label is missing.

We thank the reviewer for noticing this missing y-axis label. We have made this correction.

20. L269-279. I am surprised and puzzled by the failure of electrical stimulation to influence the rate of photo bleaching. In addition to spillover the authors may wish to explicitly mention why this does not support the idea that receptors involved in spontaneous and evoked release are the same. It seems unlikely that spillover from spontaneous release during this brief time is adequate to activate all of the receptors involved in evoked release. Additional experiments to clarify this finding seem key. Would use of high bath concentrations of calcium increase evoked release sufficiently so that brief periods of stimulation could be tested. Presumably evoked release would be increased proportionately more than spontaneous release, so that less bleaching may occur in the control experiment sounds stimulation.

We thank the reviewer for this insightful comment. One explanation as to why electrical stimulation during photobleaching did not further increase the bleaching of evoked responses is the low release probability of hippocampal synapses. Many release sites were likely not activated during the stimulation and thus could not be bleached. To test this idea, we increased the probability of release by increasing extracellular ca^2+^ concentration to 8 mM. In these conditions, detection of evoked responses was photobleached more when stimulation was applied during photobleaching (these results are shown in the new Supp. Figure 2I-J), thus indicating that electrical stimulation does influence the rate of photobleaching indeed.

Reviewer #2 (Recommendations for the authors):Overall, this is an exciting and potentially very impactful study. The work presented is of very high quality and there is great interest in understanding spontaneous vs evoked release. Because this study is extremely technical in nature, there are a number of technical and methodological points that need some additional clarification and consideration to properly interpret the findings.– An interesting and central finding of this study is that iGluSnFR appears to be concentrated at synapses. This seems counter intuitive. Synaptic real estate is so in demand and many molecules have specific targeting mechanisms to bring them to the synapse. How does iGluSnFR end up concentrated at the synapses without a specific synapse targeting mechanism? Is this finding solely based on imaging data gathered during high frequency stimulation? If so, one could argue that GluSnFR is not enriched in synapses but is just highly active during intense neuronal activity. Some clarification on GluSnFR protein vs GluSnFR signal localization would improve this argument.

The STORM imaging was done on fixed hippocampal cultures. To identify iGluSnFR, we immunolabled the GFP protein that is a part of the iGluSnFR molecule to stain them for STORM imaging. Thus, we are not using the fluorescence of iGluSnFR itself but rather labeling all of the iGluSnFR proteins, and not just the ones that are near glutamate release sites. We apologize for the confusion in our initial explanation and hope the improved language in the revised manuscript is more clear. As for why we might see iGluSnFR at a more concentrated level at synapses, our findings suggest that the mechanical trapping of those very synaptic molecules may actually increase the density of iGluSnFR.

– Second similar question – does the synaptic enrichment of GluSnFR fully recover after photobleaching and if so on what timescale? Because GluSnFR uses a DF/F approach to detect glutamate, it would be helpful in interpreting the evoked vs spontaneous FRAP recovery data to see the recovery of basal fluorescence (as in Figure 1C, expressed in AU) vs the ability to detect glutamate (as in the rest of the figures, expressed in DF/F).

The reviewer raises an interesting question. Synaptic enrichment was observed via STORM and seems to be independent of the activation of iGluSnFR by glutamate. The FRAP experiments on live cells suggest that iGluSnFR are trapped at evoked subsynaptic sites, but they do not provide information as to whether the synaptic enrichment is lost. Based on our experiments we speculate that synaptic enrichment of proteins in general is not altered (since electrophysiological parameters remain unchanged).

Whether the basal fluorescence changes as a result of photobleaching is an interesting question. We analyzed how the pre-photobleaching baseline fluorescence compares to the post-photobleaching baseline fluorescence. We found that while baseline fluorescence is significantly decreased from baseline after 10-20 minutes of photobleaching, there is no difference of stimulating versus not stimulating while photobleaching on the baseline fluorescence. (Supp. Figure 3J)

– How was the mobile vs immobile fraction of iGluSnFR calculated? It's not clear from the text exactly how this ratio was arrived at.

We have added a more detailed explanation in the Results and Methods sections, as well as a cartoon in Figure 4B exemplifying the analysis of FRAP experiments. We used a free web-based software that has been widely validated by other labs (see Koulouras et al., 2018).

– The photobleaching experiments used a 1 μm diameter photobleaching area. ROIs used for quantifying GluSnRF activity in the synapse used a 2-3 μm diameter ROI. If the photobleaching was centered over the active zone of the synapse, it stands to reason that the region associated with evoked release (central within the synapse) and the region associated with spontaneous release (peripheral within the synapse) are not equally photobleached. That could explain some of the data and should be considered as a potential caveat. Have the authors considered using a larger area of photobleaching to completely bleach everything in the synaptic ROI?

The FRAP experiments shown in Figure 4 were performed using a confocal microscope, and only the fluorescence changes inside the bleached ROI were analyzed (although a neighboring unbleached region was used as control for normalization). We did not do any analysis of evoked or spontaneous neurotransmission combined with FRAP. The experiments shown in Figure 5 were performed in an epifluorescence microscope, since this allows for the monitoring of multiple synapses (50-100) with low phototoxicity (since the experiments analyzing neurotransmission are long). The whole field of view was photobleached, which allowed the analysis of spontaneous and evoked neurotransmission at the same dendritic spines before and after bleaching. For the experiments analyzing neurotransmission, the subsynaptic regions responding to evoked and spontaneous release are equally photobleached. We have more clearly explained these differences in the revised manuscript.

– In Figure 1 C it would be helpful to show PSD95. The figure legend says that the particles shown colocalize with PSD95 but there is no way to interpret or evaluate that with the current figure presentation.

We have included STORM images of PSD95 in the new Supp. Figure 1 and discussed this in the revised Results section.

– Because the central focus of the paper is spontaneous synaptic release, it would help the manuscript to give the reader a better description of exactly how spontaneous release events were detected. In the paper the authors state "we examined parameters of spontaneous iGluSnFR events" but it's not clear how they got those parameters to evaluate. Were the exact same ROIs used? Was there a signal to noise threshold used to identify events from noise? Were spontaneous events captured before or after evoked? Was that switched to control for the potential effects of stimulation of spontaneous release?

The same ROIs were used for evoked and spontaneous release analysis. The SNR threshold was 3 SD above baseline for both evoked and spontaneous. Spontaneous events were captured before evoked to avoid potential confounding effects of stimulation, as shown in Figure 5A. We stimulated at a frequency of 0.2 Hz, which is not sufficient to induce plasticity. We also performed control experiments in which we measured evoked release before spontaneous and it did not affect the results (data not shown). These details have been added to the Results and Methods sections of the revised manuscript.

– It is not really clear why the authors did not use TTX throughout their experiments to analyze spontaneous release. Using TTX is the standard in the field for isolating action-potential independent events. I see that it allows them to do their stimulation experiments in the same cultures as their evoked experiments, which is essential to their study, but it leaves open the possibility that some spontaneous events are AP-associated. Have the authors made recordings from neurons in glutamate synaptic blockers and shown that 100% of the AP are absent? That would be a helpful control to validate their method.

We tested whether the spontaneous event frequency in cultures with CNQX and APV was different to CNQX, APV and TTX. We found no significant difference in these two groups, suggesting that there is no significant contribution of action potentials when CNQX and APV are added in the imaging solution (Figure 3F). Furthermore, in earlier work our group has analyzed the frequency of spontaneous event as detected by presynaptic vGlut1-pHluorin fluorescence (Leitz and Kavalali, 2014) or postsynaptic GCAMP6-PSD95 fluorescence (Reese and Kavalali, 2016) in the presence of TTX or CNQX (also AP5 is some cases). These experiments showed that suppression of network activity with CNQX is sufficient to block AP firing and thus isolate genuine spontaneous events.

– GluSnFR detection of evoked release is suppressed for hours after photobleaching while the ability to detect spontaneous release using GluSnFR recovers much more rapidly. This is a very clear and exciting finding! An implication of this result, however, is that 100% of the events that contribute to evoked release are spatially compartmentalized from any of the machinery that drives the ability to detect spontaneous release. Spontaneous release is detected using the same ROIs as evoked release (as far as I can tell), so that means within an ROI spontaneous release is being detected while evoked is not. Data presented shows that glutamate release is happening in both situations, but just can't be detected in the evoked release situation due to the photobleaching. So evoked release is somehow spatially restricted to the GluSnFR molecules in the synapse and cannot be detected by other GluSnFR molecules that detect spontaneous release in the same ROI. This is hard to reconcile with the fact that glutamate diffusion is likely the driving factor by which glutamate is cleared from the center of synapses.

We thank the reviewer for the insightful comment. We agree, our interpretation is also that the postsynaptic regions that respond to evoked glutamate release are compartmentalized, i.e. spatially segregated from spontaneous, and have less lateral mobility.

Regarding glutamate diffusion, diffusion of glutamate at the synaptic cleft has been proposed to be slower than in solution or in non-synaptic regions (~0.5-0.7 µm^2^/ms). These values still lead to diffusion outside the synapse in just a few milliseconds, consistent with electrophysiological and optical measurements. However, the glutamate concentration nanodomain at the cleft seems to be extremely narrow and short lived. Glutamate can reach a concentration close to 1 mM for ~50-100 µs in a small 100-250 nm region at the cleft, and concentration drastically decays laterally. This model agrees with the experimental observation that postsynaptic glutamate receptors are not saturated by single AP stimulations, and would support the nanodomains or subsynaptic compartmentalization of signaling at synapses. Taken together, these previous antecedents are not contradictory, but rather agree with our current observations.

Bibliography used: Zheng et al., Scientific Reports 2017; Budisantoso et al., J. Physiol. 2013; Clemens et al., Science 1992; Raghavachari and Lisman, JNP 2004; Savtchenko and Rusakov, Philos Trans R Soc Lond B Biol Sci. 2013.

Reviewer #3 (Recommendations for the authors):This paper by Wang CS et al., investigates the localization of spontaneous and evoked vesicle fusion within a synapse. This is an important question and an interesting central finding. The authors nicely document the use of a glutamate-sensitive fluorescent reporter (iGluSnFR) to measure spontaneous and evoked vesicle fusion. They do a great job of characterizing iGluSnFR and take advantage of the probe's rapid bleaching to follow up previous molecular and physiological characterization of spontaneous vesicle fusion with novel findings of unique spatial segregation of spontaneous vesicle fusion. This finding is mainly based on this bleaching characteristic of GluSnFR. I was impressed by the idea of using the bleaching of a membrane probe to resolve the location of vesicle fusion. I have some questions and comments that I believe are important to be addressed to better understand the findings and their interpretation for an accepted manuscript.1. iGluSnFR has a very high affinity to provide it with the sensitivity for detecting single vesicle fusion. A concern would be if the sensor is also detecting vesicle fusion as spillover from adjacent boutons of untransfected neurons undergoing spontaneous release. This concern is somewhat mitigated since the rate of spontaneous release is not much higher than expected from electrophysiology and good criteria are deployed for detection of events vs noise. However given that location is so important for the interpretation, these "spill-over" events could appear to be released from unique discrete areas of the synaptic bouton but are in fact coming from nearby synapses and the diffusing glutamate is being detected. A better characterization of the location of where spontaneous events were detected optically in the synapse relative to evoked might help resolve this concern. Were the ROIs formed from high frequency stimulation (or high potassium?) also in the same location for measuring spontaneous release? Were spontaneous vesicle fusions also detected outside of synapses identified by stimulation that might suggest spillover or solely restricted to regions with evoked release identified on dendrites?

Yes. the same ROIs were used for evoked and spontaneous release measurements and thus evoked and spontaneous neurotransmission was monitored in the same location. We did not draw ROIs on dendritic shafts or cell bodies, only dendritic spines. We have clarified these points in the revised manuscript.

2. The authors lead me to believe that GluSnFR bleaching is primarily dependent on activation of the indicator by glutamate binding. The difference in bleaching from glutamate perfusion compared to during rest or "no stimulation" for selectively impairing detection of evoked vs spontaneous release is striking and really interesting. That being said, it is very hard to understand the explanation the authors provide for the lack of difference between unstimulated and stimulated bleaching conditions. A difference here seems like the most direct experimental test to define separate release sites for evoked and spontaneous. If I understand the authors correctly, the lack of difference is the result of the spontaneous release of glutamate that contaminates this result. However, spontaneous release has a very low rate and would be releasing very small amounts of glutamate over the bleaching times identified. During 2 minutes of photobleaching when a very acute difference is found the rate of 0.1 Hz for spontaneous release the authors measured would mean that synapses experience on average 1 spontaneous vesicle fusion event during 2 minutes of photobleaching. This one release event would not be localized to sites of evoked release as the authors interpret the data yet would cause substantial bleaching equivalent to several vesicles fusing during stimulation. As it stands, 30 APs delivered during standard imaging over 2.5 minutes don't seem to cause any bleaching or decrease in detection efficiency (Figure 5A). To better understand the amount of bleaching that occurs in the three conditions, it would be useful to know both the difference in the illumination intensity between standard measurements and during photobleaching as well as the changes in resting fluorescence iat the synapse/ROIs. Presumably at the bleaching laser illumination power the same delivery of 30 APs @ 0.2 Hz (2.5 minutes) would rapidly decrease to very low levels if this interpretation of rapid bleaching when glutamate in present is correct. Otherwise, bleaching of GluSnFR doesn't require glutamate and perhaps another unknown property of local GluSnFR diffusion within a spine that the authors mention is responsible for the lack of difference in bleaching sensitivity to boutons at rest vs stimulation. This could still be in line with the authors' interpretation of unique release sites for evoked neurotransmission, but less easy to understand than selective loss from use of evoked release sites by glutamate dependent photobleaching. Perhaps lowering the "bleaching" laser intensity during stimulation would ablate/bleach detection of evoked release during stimulation that also preserves spontaneous release detection.

We thank the reviewer for this insightful comment. We think that one possible explanation for the lack of difference between unstimulated and stimulated bleaching conditions is the low release probability of hippocampal synapses. Many release sites were probably not activated during the stimulation, which is why they could not be bleached. To test this hypothesis, we raised bath ca^2+^ concentration to 8 mM to increase release probability. In high ca^2+^, iGluSnFR evoked responses were more highly reduced in the stimulated photobleaching condition compared to the unstimulated one (these results are shown in the new Supp. Figure 2I), thus indicating that electrical stimulation does influence the rate of photo bleaching and that evoked subsynaptic sites may be segregated from spontaneous ones.

Between imaging and photobleaching, we use different light intensities. During photobleaching, we remove the neutral density filter within the Λ-DG4 illumination system to increase the emission light intensity to 100%. We have included this information in the revised manuscript.

Finally, we analyzed the baseline fluorescence before and after photobleaching as well as with and without stimulation. While we see that baseline fluorescence is affected by photobleaching, indicating that tonically and spontaneously activated iGluSnFR is also bleached, stimulating during the photobleaching does not differentially affect this decrease (new Supp. Figure 3J).

[Editors’ note: further revisions were suggested prior to acceptance, as described below.]

Reviewer #1 (Recommendations for the authors):This resubmission describes work using a genetically encoded glutamate sniffer to study synaptic transmission in hippocampal cultured neurons. It investigates the properties of evoked and spontaneous transmission and identifies differences in the rates of photobleaching suggesting that the receptors involved in these 2 signaling pathways are localized differently. The authors have made a number of changes to the initial submission that have improved the manuscript, however, there are a number of remaining problems that need to be addressed before it is suitable for publication in Life.Although much improved, the manuscript may be easier to read if the authors began the Results section by describing the distribution of iGluSnFR and how this compares with that for vGlut1 and PSD 95 distribution to indicate that the glutamate sniffer is found near to synapses (Figure 1 Suppl 2). They could then move on to describing iGluSnFR localization via super-resolution microscopy and then show how excitatory puncta number are similarly impacted by transfection with GFP, the sniffer, and MAP2.

We have rearranged this section of the results to improve the flow as suggested by reviewer.

The authors should also improve the description of their approach to analyzing the experiments that employed super-resolution microscopy. They describe experiments in which PSD 95 and GFP staining is identified in fixed issue and how they use the DB scan method to identify that this staining is clustered for both PSD and GFP. Some of these sites colocalize. They then go on to use pair correlation to examine the clustering of the iGluSNFR signal and show that it is not uniformly distributed within each of the clusters but highly concentrated close to its centre. It is unclear to me what criteria the authors use to justify describing this as the centre of the synapse (line 111).

The program identifies the geometrical center of the cluster to run a pair correlation analysis. We have included this detail in the methods section of the paper.

In an earlier review, (question 5) I asked to see a control image from before stimulation of old Figure 1B. The summation projection is now in figure 2A but there is still no control image to illustrate how the fluorescence staining of iGluSNFR increases. Perhaps the authors have a control image for this, or another similar experiment, that could be included in this figure or a supplementary figure.

We have included a control image to demonstrate increase in fluorescence staining of iGluSnFR

On line 155 of the manuscript the authors describe using regions of interest of 2 µm in diameter. However, in the methods section they mention ROIs were 2 to 3 µm in diameter. Please clarify if there is a discrepancy or the methods are referring to different experiments. Also, please mention if the fluorescence signal captured from the region of interest the sum, average or maximum values from the pixels therein?

ROIs were drawn with 2 µm diameter size; we have updated the methods section.

Line 222, can the authors provide a citation confirming that the antagonists CNQX and APV do not interact with iGluSNFR?

We now include a citation of the original Looger paper from 2013, which demonstrates that different pharmacological inhibitors including CNQX and APV do not have any detectable affinity for iGluSnFR.

Line 287, can the authors confirm that it is only in the experiments testing probe mobility that they used a 1 µm diameter region of interest for bleaching purposes. In all later experiments, using bleaching as a tool did they photobleach the entire field of view and were these same experiments performed using an EM CCD. Please clarify these details in the manuscript. Similarly, is it the case that in the late experiments shown in figure 5 and associated supplementary figure, that the regions of interest were always identified by activity during the evoked phase of the experiment, or were some regions of interest identified during the spontaneous phase? This information could be confirmed in the text to emphasize to the reader that the two segregated detection systems for spontaneous and evoked release are tightly associated.

We confirm that this was the experimental parameters performed, and we clarify this in the manuscript. In the experiments associated with figure 5 and its supplementary figure, regions of interest were identified during the evoked phase similar to experiments performed for figures 1 and 2. We have included these details in the manuscript.

Rather surprisingly the authors find the rate of bleaching of evoked signals is unchanged by stimulation during the bleaching. Given that there is no iGluSNFR response, I do not understand the hypothesis advanced concerning low probability of release and unbleached responsive sites mentioned on lines 356-358. The data appear to show that bleaching was saturated after a few minutes of photobleaching with no later responses to stimulation as mentioned in the hypothesis.

We have edited the text to improve the clarity of this explanation.

However, I think the experiments using high calcium follow-up are interesting. It seems that the high calcium is reducing the photobleaching in the lack of stimulation compared to the low calcium experiments (Figure 5 supp 1 G). I wonder if the results could also be explained by reduced action potential firing due to reduced excitability in the presence of high calcium. In addition, the bleaching is accelerated in the spontaneous release experiments using high calcium. Presumably this is because of increased rates of spontaneous release. These possibilities should be included in the discussion.

We have included the possibility of reduced action potential firing in the presence of high calcium. We did not include the effects of 8mM Ca on spontaneous release photobleaching in the supplemental figure.

These differences reported in the rate of recovery of glutamate signaling, vividly underline the authors' thesis that movement of iGluSF was differentially impacted at sites that detect spontaneous and evoked release. Their major result showing that detection of evoked and spontaneous release detected in a 2 µm region of interest are so differently affected following bleaching, supports the authors' ideas that there is separation between the postsynaptic pathways mediating evoked and spontaneous release.Have the authors estimated what fraction of excitatory synapses in their cultures express the iGluSnFR and how much of the iGluSnFR are not clustered at synapses? The first part of this question may be of interest to others and the second part may help identify the source for the iGluSnFR that reverses the bleaching as the neurons recover. If there was a wide field of bleaching and most of the iGluSnFR is clustered at synapses then one imagines that there is little expressed iGluSnFR to diffuse to the affected areas and maybe new protein formation may be required. Is it possible that the iGluSnFR at evoked sites is more sensitive to bleaching compared to that involved in mini detection and that contributes to the differences observed?

We have done these estimations before; however, there is clearly an extrasynaptic pool as indicated in the representative image in figure 2 showing baseline fluorescence at the plasma membrane due to activation from basal levels of extracellular glutamate. Furthermore, the version of iGluSnFR we use is not targeted and thus would be expressed across the entire plasma membrane. Thus, it is expected that there is a large amount of iGluSnFR not clustered at synapses. Furthermore, we do not necessarily claim nor know that every synaptic site contains a detectable cluster of iGluSnFR.

Indeed, iGluSnFR at evoked sites is more sensitive to photobleaching compared to spontaneous bleaching, and this is because the iGluSnFR at evoked release sites does not have as much access to the extrasynaptic pool as spontaneous release sites.

Reviewer #3 (Recommendations for the authors):As before, the findings of this manuscript are very interesting. The authors addressed many of my concerns with the initial manuscript and have now included the necessary experimental details. I believe that the measurements and experiments conducted in this paper can now be made by other groups and the results understood to support the conclusions of the authors. I believe that supplementary experiments throughout are useful.Figure 5 still seems to me to be a critical experiment for the interpretation and I found two parts a little unclear in the revisions, and could benefit from additional edits.Line 354-355: "One explanation is basal activity…." What do you exactly mean by basal activity? Does this mean spontaneous release or action potentials and evoked release occurring independently of your stimulation protocol? From the methods I was under the impression that CNQX and APV were present to block electrical activity outside of field stimulation, so I wanted to be sure you were suggesting spontaneous release, but expansion here would be helpful to know exactly what the authors mean.

We meant spontaneous release (not triggered by action potentials), and we have included this phrase to improve the clarity of this sentence.

Line 359-360"To test the latter hypothesis, we repeated the experiments in a higher extracellularca^2+^ of 8 mM. In this condition, probability of release is close to 1, and most synaptic sites should be activated and photobleached."It would be more helpful to include the exact changes in Pr from the paper in these calcium conditions, shifting from (~ 0.5 and 0.9). I could not find the exact number in Figure 2 or the manuscript but estimated from the bar graphs.

We have included the exact number in the Results section that correspond to Figure 2.

Otherwise, the changes were very helpful and made the manuscript more compelling.